# Osteoclasts protect bone blood vessels against senescence through the angiogenin/plexin-B2 axis

Xiaonan Liu [1,2,5], Yu Chai [1,2,5], Guanqiao Liu [1,2], Weiping Su[1], Qiaoyue Guo[1], Xiao Lv [1], Peisong Gao [3], Bin Yu[2], Gerardo Ferbeyre [4], Xu Cao [1] & Mei Wan [1✉]

Synthetic glucocorticoids (GCs), one of the most effective treatments for chronic inflammatory and autoimmune conditions in children, have adverse effects on the growing skeleton. GCs inhibit angiogenesis in growing bone, but the underlying mechanisms remain unclear. Here, we show that GC treatment in young mice induces vascular endothelial cell senescence in metaphysis of long bone, and that inhibition of endothelial cell senescence improves GC-impaired bone angiogenesis with coupled osteogenesis. We identify angiogenin (ANG), a ribonuclease with pro-angiogenic activity, secreted by osteoclasts as a key factor for protecting the neighboring vascular cells against senescence. ANG maintains the proliferative activity of endothelial cells through plexin-B2 (PLXNB2)-mediated transcription of ribosomal RNA (rRNA). GC treatment inhibits ANG production by suppressing osteoclast formation in metaphysis, resulting in impaired endothelial cell rRNA transcription and subsequent cellular senescence. These findings reveal the role of metaphyseal blood vessel senescence in mediating the action of GCs on growing skeleton and establish the ANG/PLXNB2 axis as a molecular basis for the osteoclast-vascular interplay in skeletal angiogenesis.

[1] Department of Orthopaedic Surgery, The Johns Hopkins University School of Medicine, Baltimore, MD, USA. [2] Division of Orthopaedics and Traumatology, Department of Orthopaedics, Nanfang Hospital, Southern Medical University, Guangzhou, Guangdong, China. [3] Johns Hopkins Asthma & Allergy Center, Johns Hopkins University School of Medicine, Baltimore, MD, USA. [4] Department of Biochemistry and Molecular Medicine, Université de Montréal, Montreal, QC, Canada. [5] These authors contributed equally: Xiaonan Liu, Yu Chai. ✉email: mwan4@jhmi.edu

Synthetic glucocorticoids (GCs) are one of the most effective treatments for a wide range of chronic inflammatory, autoimmune, and neoplastic diseases. Approximately 10% of children receive GC treatment at some point during childhood[1], and long-term GC treatment is common in the management of many chronic childhood illnesses[2–7]. The use of GCs has improved outcomes and survival rates for children with these disorders; however, the adverse effects of GCs on bone can be severe. GC-induced osteoporosis affects 30–50% of patients treated with long-term GC therapy, including children[8–10]. Because bone grows rapidly in healthy children, their skeletons are particularly vulnerable to the adverse effects of GCs on bone formation. GC-induced skeletal disorders in children include growth retardation, impaired acquisition of peak bone mass, early-onset osteoporosis with a high fracture rate, and osteonecrosis[11–13]. Impaired acquisition of peak bone mass or loss of bone mass during childhood has lifelong implications, including greater fracture risk throughout adulthood[14–16]. There are still no ideal medications to treat osteoporosis in children[17]. The mechanisms by which GCs suppress the acquisition of bone mass need to be further elucidated.

Studies conducted during the past 2 decades have shown that the adverse effects of GCs on the skeleton result from both systemic and local actions on many types of skeletal cells including osteoclasts[8,12,18–22], osteoblasts[12,23,24], osteoprogenitor cells[25,26] osteocytes[27–29], and chondrocytes at growth plates[22,30–33]. The effect of GCs on bone blood vessels was previously noted. GC treatment led to disruption of the vascularity of the femoral head[34]. During the postnatal growth period, GCs prevent blood vessel invasion (i.e., angiogenesis) at the chondro-osseous junction between metaphyseal bone and the growth plate, thereby inhibiting the formation of primary spongiosa[35,36]. The vascular system is indispensable for bone health because osteogenesis requires an adequate blood supply. During development and postnatal growth of the mammalian skeletal system, bone formation is tightly coupled with angiogenic growth of blood vessels[37,38]. In addition, the local vasculature provides a niche for mesenchymal stem/progenitor cells and thereby regulates osteogenesis[39–42]. However, it remains largely unknown on how GCs negatively regulate angiogenesis in postnatal growing bone.

Cellular senescence represents a series of diverse, dynamic, and heterogeneous cellular states with irreversible cell-cycle arrest, apoptosis resistance, and the senescence-associated secretory phenotype[43–50]. Cellular senescence can be induced by many stimuli, including but not limited to, the end of replicative lifespan/telomere erosion, DNA damage, mechanical stress, and oncogenic stimuli. Senescent cells exhibit essentially stable cell-cycle arrest through the actions of tumor suppressors such as p16INK4a, p15INK4b, p53, p21CIP1, and others[44,47]. Senescent cells also have increased activity of senescence-associated β-galactosidase (SA-βGal) and loss of nuclear expression of high mobility group box 1 (HMGB1)[51]. Ribosome biogenesis defect is also an important contributor to cellular senescence. Both the p53 and the RB tumor suppressor pathways are activated by ribosome-free ribosomal proteins that accumulate in cells when the maturation or synthesis of ribosomal RNA (rRNA) is interrupted. The 5S ribonucleoprotein (RNP) components are involved in p53 activation[52], whereas the ribosomal proteins RPS14 or uS11 and RPL22 act as inhibitors of the cyclin-dependent kinase (CDK)4/6 inhibitor[53,54]. A recent study demonstrated that inhibition of the 60S ribosome biogenesis led to a robust induction of cellular senescence[55]. Furthermore, Nol12, an rRNA-binding protein, is associated with nucleolar stress-driven cellular senescence and normal aging[56].

Angiogenin (ANG) is a secreted ribonuclease with potent growth-promoting activity[57–59]. Particularly, ANG stimulates endothelial cell growth and proliferation, mainly by promoting 47S rRNA transcription[60–62]. Recently, plexin-B2 (PLXNB2), a transmembrane receptor belonging to the large family of plexin proteins, was identified as a functional ANG receptor that is both necessary and sufficient for the physiological and pathological functions of ANG in multiple cell types, including endothelial, cancer, neuronal, normal hematopoietic and leukemic stem and progenitor cells, as well as intestinal epithelial cells[63,64]. Upon secretion, ANG is endocytosed in target cells via PLXNB2[62,65] and accumulates in the nucleus, where it regulates rRNA transcription to promote cell proliferation[61,66–69]. When nuclear translocation is inhibited, its angiogenic activity is abolished[70].

In the current study, we show that synthetic GC treatment in growing young mice induces cellular senescence of blood vessel endothelial cells in the metaphyseal region of long bone, resulting in diminished angiogenesis and coupled osteogenesis. Mechanistically, osteoclast-derived ANG maintains osteogenesis-coupled type-H vessels in the metaphysis of growing skeleton via an ANG/PLXNB2-ribosomal biogenesis signaling pathway; Inhibition of this signaling pathway by GCs leads to vascular cell senescence and resultant impaired angiogenesis-osteogenesis. Finally, we demonstrate that antagonizing cellular senescence by delivering recombinant human ANG (rhANG) in vascular endothelial cells blunts the adverse effects of GCs on the growing skeleton.

## Results

### Senescent cells accumulate in metaphysis of long bone in young mice after GC treatment

In our previous work, we recognized cellular senescence in primary spongiosa of long bone during late puberty as a normal programmed process, representing an important signature of skeleton for the transition from fast growth to slow growth[71]. Given that GCs impair bone growth and mineral acquisition in growing bone, we investigated whether GC treatment in prepubertal or early pubertal mice induces aberrant cellular senescence, which leads to bone growth cessation. Here, we used a murine senescence reporter strain p16tdTom (Fig. 1a), in which tandem-dimer Tomato (tdTom) was knocked into exon 1α of the p16INK4a locus to enable the identification of p16INK4a-activated cells (tdTom+) at the single-cell level[72]. Daily injection of methylprednisolone (MPS), a commonly used synthetic GC, into 3-week-old (prepubertal) p16tdTom mice for 2 weeks significantly increased the frequency of tdTom+ cells derived from the whole metaphysis of the femoral bones as detected by flow cytometry (Fig. 1a–c), indicating high-level activation of the p16INK4a promoter. Consistently, in situ fluorescence analysis of the femoral bone tissue sections revealed a much greater number of tdTom+ cells in both primary and secondary spongiosa regions in MPS-treated mice compared with vehicle-treated mice (Fig. 1d–f). We also conducted SA-βGal staining using the bone tissue sections and found an increase in the number of SA-βGal+cells in primary and secondary spongiosa regions in MPS-treated mice relative to vehicle-treated mice (Fig. 1g–i). Increased numbers of SA-βGal+ cells were not detected in the diaphyseal bone marrow in MPS-treated mice relative to vehicle-treated mice (Supplementary Fig. 1). Immunofluorescence staining showed that whereas nuclear localization of HMGB1 was seen in most of the cells in metaphysis of vehicle-treated mice, many cells exhibited relocalization of HMGB1 from the nucleus to the cytoplasm and an overall reduced fluorescence intensity in MPS-treated mice (Fig. 1j–l). Therefore, GC treatment induces cellular senescence in the metaphysis of growing bone.

### Metaphyseal vascular cells and osteoprogenitors undergo senescence in response to GC treatment

It is well-recognized that angiogenesis coupled with osteogenesis is abundant in the

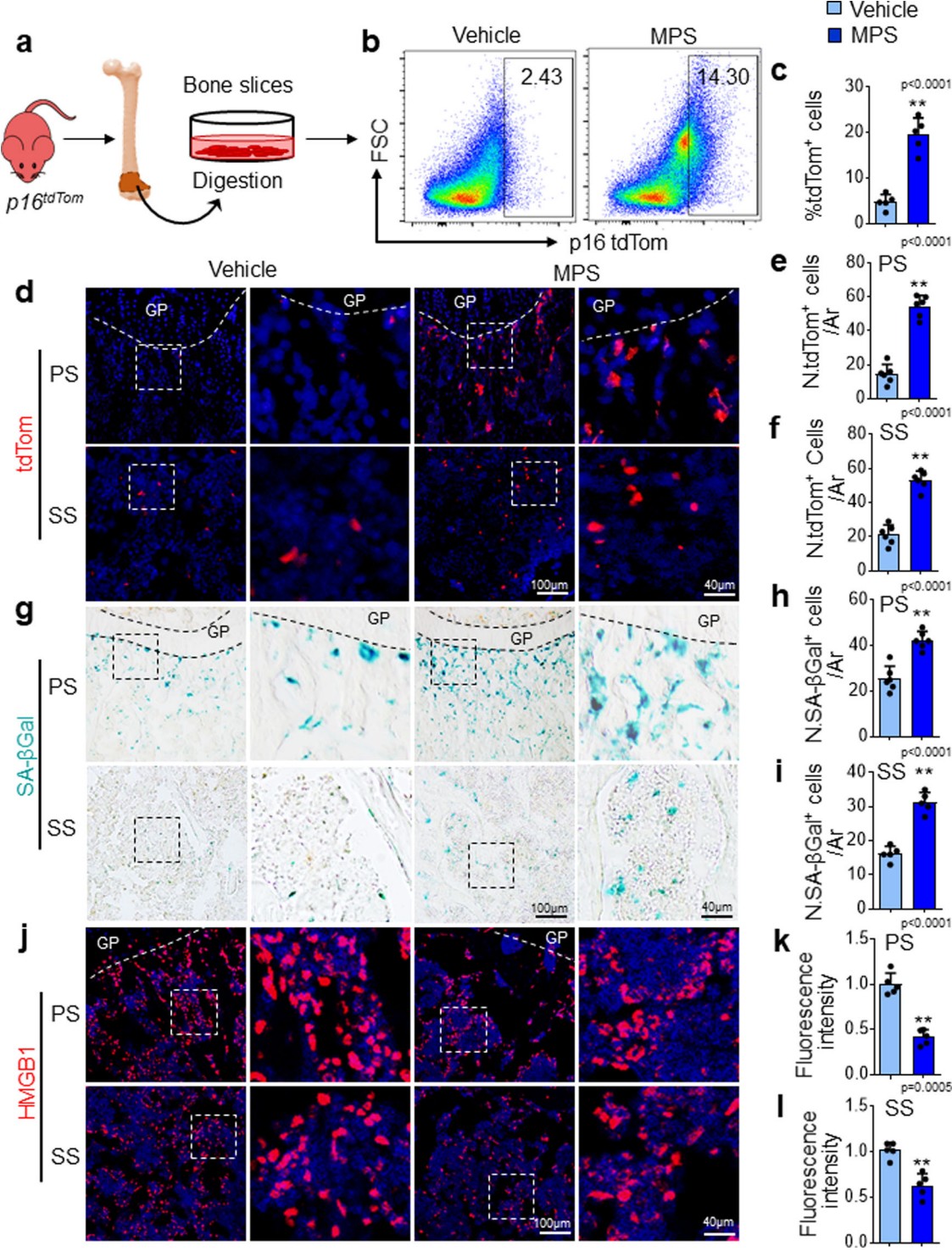

metaphysis of the growing skeleton in mice but declines in young adult mice, in which bone growth slows[71,73–75]. Therefore, it is logical to postulate that highly proliferative vascular cells and osteoprogenitor cells in metaphysis are particularly vulnerable to GC treatment. Indeed, co-staining of bone tissue sections with tdTom and a specific vascular endothelial surface marker, endomucin (Emcn), showed increased numbers of tdTom-expressing vascular endothelial cells in metaphysis in MPS-treated vs. vehicle-treated mice, starting at 1 week, reaching a peak at 2 weeks, and declining at 3 weeks after treatment (Fig. 2a and b). The reduced senescent cells at 3-week relative to 2-week MPS

treatment are likely because of the senescent cell clearance by immune cells. The number of osterix (Osx)+ osteoprogenitor cells that expressed tdTom also increased in response to MPS but maintained a similar level at 1–3 weeks after treatment (Fig. 2c, d). Of note, the number of tdTom-expressing cells increased nearly 6-fold in the Emcn+ cell population but only to 3-fold in the Osx+ cell population at the peak (Fig. 2b and d), suggesting that vascular endothelial cells are more susceptible than osteoprogenitor cells to MPS treatment. Consistently, flow cytometric analysis of the cells in metaphysis also showed a strong induction of tdTom expression in CD144+ cells (including vascular endothelial cells and

**Fig. 1 Senescent cells accumulate in metaphysis of long bone in young mice after GC treatment. a–c** Flow cytometry analysis of the tdTom$^+$ cells in femoral metaphysis. **a** Schematic diagram illustrating the experimental procedure. Three-week-old $p16^{tdTom}$ mice were treated with methylprednisolone (MPS) at 10 mg/m$^2$/day or vehicle by daily intraperitoneal injection for 2 weeks. Metaphyseal bone tissue from distal femur was digested, and the isolated cells were subjected to flow cytometry analysis (see the detailed description in the "Methods" section). Representative images of tdTom-expressing cells of the femoral metaphysis is shown in (**b**). Percentages of tdTom$^+$ cells in bone/bone marrow are shown in (**c**). **d–f** Three-week-old $p16^{tdTom}$ mice were treated with MPS at 10 mg/m$^2$/day or vehicle by daily intraperitoneal injection for 2 weeks. Representative confocal images from frozen sections of the femur in (**d**). Red: tdTom$^+$ cells; Blue: nuclear staining by DAPI. Boxed areas are shown at a higher magnification in corresponding panels to the right. Quantified numbers of tdTom$^+$ cells in primary spongiosa and secondary spongiosa per mm$^2$ tissue area (N. tdTom$^+$ cells/ Ar) are shown in (**e**) and (**f**), respectively. **g–l** Three-week-old $BALB/c$ mice were treated with MPS at 10 mg/m$^2$/day or vehicle by daily intraperitoneal injection for 3 weeks. SA-βGal staining of femoral bone sections was performed. Representative images of SA-βGal$^+$ cells (blue) in metaphysis are shown in (**g**). Quantified numbers of SA-βGal$^+$ cells in primary spongiosa and secondary spongiosa per mm$^2$ tissue area (N. SA-βGal$^+$ cells/Ar) are shown in (**h**) and (**i**), respectively. Immunofluorescence staining of femoral bone sections was performed using antibody against HMGB1. Representative images of HMGB1$^+$ cells (red) are shown in (**j**). DAPI stains nuclei blue. Quantified fluorescence intensity of HMGB1$^+$ cells in primary spongiosa and secondary spongiosa was shown in (**k**) and (**l**), respectively. GP growth plate. Ar tissue area. PS primary spongiosa, SS secondary spongiosa. $n = 5$–6 mice. Data are represented as mean ± s.e.m. **$p < 0.01$, as determined by two-tailed Student's $t$-tests.

endothelial progenitor cells)[76] in $p16^{tdTom}$ mice after MPS treatment (Fig. 2e and f) although the percentages of CD144$^+$ and Emcn$^+$ endothelial cells were decreased in response to MPS treatment (Supplementary Fig. 2). We further evaluated the senescence of endothelial cells and osteoclasts in metaphysis using another senescence marker SA-βGal. Co-staining of bone tissue sections with SA-βGal and Emcn also showed markedly increased percentage of SA-βGal-expressing blood vessels in metaphysis in MPS-treated mice relative to vehicle-treated mice (Supplementary Fig. 3a, b). While tartrate-resistant acid phosphatase (TRAP)/SA-βGal co-staining showed that ~7.67 ± 1.45% of the TRAP$^+$ osteoclasts were positive for senescence marker SA-βGal, the percentage of senescent TRAP$^+$ cells did not change in MPS-treated mice relative to vehicle-treated mice (Supplementary Fig. 4a, b). Therefore, MPS treatment does not induce cellular senescence in the osteoclast lineage in the metaphysis. GCs induce apoptosis of osteoblasts and osteocytes[28,29]. We detected whether MPS treatment also led to apoptosis of vascular cells and osteoprogenitor cells by analysis of the percentages of TUNEL$^+$ cells in different cell populations. MPS treatment indeed induced more TUNEL$^+$ cells in the metaphysis of growing bone (Supplementary Fig. 5a). Co-staining of the bone tissue sections with Emcn or Osx showed an increase in the number of TUNEL$^+$ cells in Osx$^+$ osteoprogenitor cell population (Supplementary Fig. 5c, d) but not in Emcn$^+$ vascular endothelial cells (Supplementary Fig. 5a, b) in the metaphysis of MPS-treated vs. vehicle-treated mice, suggesting that cells of osteoblastic lineage may exhibit multiple cell fate changes (apoptosis, senescence, or others) in response to GC treatment.

To further confirm the senescence of vascular cells in response to MPS treatment, we isolated CD144$^+$ cells from the femoral metaphysis (including primary and secondary spongiosa) using an established enzymatic digestion approach[71] and incubated the cells with a fluorescence βGal probe[77,78] (Fig. 2g). Flow cytometry analysis of the isolated cells showed that the total number of βGal$^+$ cells and the percentage of SA-βGal-expressing endothelial cells increased dramatically in MPS-treated mice vs. vehicle-treated mice (Fig. 2h, i). In addition, fluorescence-activated cell sorting (FACS) of the CD144$^+$ cell populations from the femoral metaphysis followed by quantitative real-time polymerase chain reaction (qRT-PCR) analysis revealed that CD144$^+$ cells from MPS-treated mice (vs. vehicle-treated mice) had reduced expression of cell proliferative marker $Ki67$ (Fig. 2j) and much higher expression of $p16INK4a$ (Fig. 2k) and $p53$ (Fig. 2l), 2 cellular senescence-inducing genes. A moderate increase in the expression of cell cycle inhibitor gene $p21CIP1$ was observed in CD144$^+$ cells from MPS-treated mice compared with those from vehicle-treated mice (Fig. 2m). We also investigated the nuclear

expression of a senescence-initiating factor, HMGB1, and a cell proliferation marker, Ki67, in single vascular endothelial cells isolated from MPS-treated and vehicle-treated mice using ImageStreamX, an imaging flow cytometer capable of producing high-resolution fluorescent images of single cells directly in flow[79–81]. We were able to identify strong HMGB1 and Ki67 signal in CD144$^+$ vascular endothelial cells isolated from vehicle-treated mice (Fig. 2n, top panel), but the positive signals for both proteins were reduced or completely lost in cells from MPS-treated mice (Fig. 2n, lower panel). Representative histograms revealed high levels of Ki67-positive (Fig. 2o) and HMGB1-positive staining (Fig. 2q) in the cells from vehicle-treated mice but dramatically reduced staining for both proteins in the cells from MPS-treated mice. Further quantification revealed that the intensity of Ki67 fluorescence in the CD144$^+$ cells were markedly reduced in the cells isolated from MPS-treated mice (Fig. 2p). Although most of the CD144$^+$ cells still expressed HMGB1 in the nucleus (Fig. 2q), the intensity of this protein decreased in mice after MPS treatment (Fig. 2r), which is consistent with previous findings that reduced HMGB1 nuclear localization initiates cellular senescence[51]. Together, the results suggest that GC treatment leads to cellular senescence of vascular endothelial cells and osteoprogenitors in metaphysis of growing bone, with the response of vascular cells to GC-induced senescence being more profound.

**Antagonizing endothelial cell senescence improve GC-impaired bone angiogenesis and osteogenesis.** We investigated the changes in CD31$^{hi}$Emcn$^{hi}$ type-H vessels, a specialized capillary subtype coupling angiogenesis and osteogenesis[73,82], in MPS-treated mice. MPS treatment induced time-dependent reductions in CD31$^{hi}$Emcn$^{hi}$ vessels in both primary and secondary spongiosa of femoral bone as detected by immunofluorescence staining (Fig. 3a–c). Consistently, flow cytometry analysis showed decreased numbers of both CD31$^{hi}$Emcn$^{hi}$ and CD31$^{lo}$Emcn$^{lo}$ cells in the same region in MPS-treated mice relative to vehicle-treated mice, with CD31$^{hi}$Emcn$^{hi}$ type-H vessels being more profoundly affected (Fig. 3d–f), suggesting that the osteogenesis-coupled type H vessels are the predominant target of GCs. The numbers of osteocalcin (OCN)$^+$ osteoblasts (Fig. 3g–i) also decreased in the metaphysis after MPS treatment in a time-dependent manner, indicating gradually diminishing osteoblastogenesis in this area. Of note, the intensity of type H vessels and numbers of OCN$^+$ cells are different at 1 and 3 weeks in vehicle-treated mice. This is because the treatment of the mice started at 3 weeks of age, and mice were euthanized at 4 and 6 weeks old, respectively, in these two treatment groups. This result agrees with the previous finding that angiogenesis and

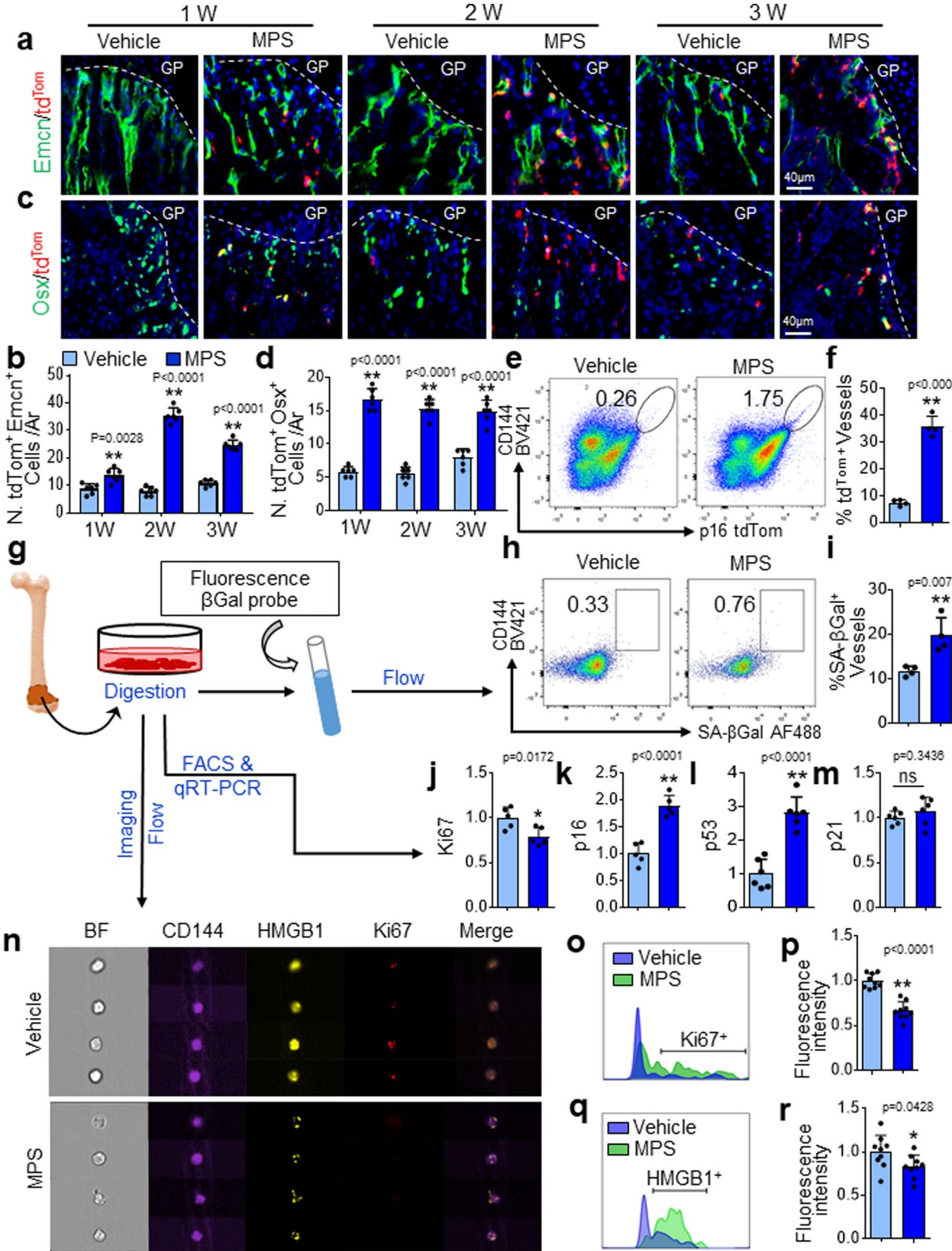

coupled osteogenesis are abundant in growing young mice and gradually decline with age[73,83].

We then examined whether deletion of a key senescence gene *p16INK4a* may rescue GC-induced defects in bone angiogenesis and osteogenesis. We evaluated the bone phenotype of *Cdh5-Cre^ERT2*; *p16^flox/flox* (*p16*-iKO) mice, in which *p16INK4a* is

deleted selectively in vascular endothelial cells in a tamoxifen-inducible manner. Endothelial cell senescence as indicated by the percentage of Emcn+ blood vessels that express SA-βGal (Fig. 3j and k), angiogenesis as indicated by the intensity of type-H vessels (Fig. 3l and m), and osteogenesis as indicated by the number of OCN+ cells (Fig. 3n and o) were unchanged in the

**Fig. 2 Metaphyseal vascular cells and osteoprogenitors undergo senescence in response to GC treatment. a–d** Three-week-old $p16^{tdTom}$ mice were treated with MPS at 10 mg/m$^2$/day or vehicle by daily intraperitoneal injection for 1–3 weeks. Immunofluorescence staining of femoral bone sections was performed using antibody against Endomucin (Emcn) and Osterix (Osx), respectively. Representative confocal images were shown in (**a**) and (**c**). Red: tdTom$^+$ cells; Green: Emcn$^+$ or Osx$^+$ cells; Blue: nuclear staining by DAPI. Quantified numbers of Emcn and tdTom double-positive cells (N. tdTom$^+$ Emcn$^+$ cells/Ar) and Osx and tdTomdouble-positive cells (N. tdTom$^+$Osx$^+$ cells/Ar) in primary spongiosa are shown in (**b**) and (**d**), respectively. **e** and **f** Three-week-old $p16^{tdTom}$ mice were treated with MPS or vehicle for 2 weeks. Bone cells were isolated from femoral metaphysis for flow cytometry analysis. Representative images are shown in (**e**). Percentages of tdTom-expressing cells in total CD144$^+$ vascular cell population are shown in (**f**). **g–i** Three-week-old $BALB/c$ mice were treated with MPS at 10 mg/m$^2$/day or vehicle by daily intraperitoneal injection for 3 weeks. Schematic diagram showing the experimental procedure (**g**). Cells were isolated from femoral metaphyseal region, and the cell suspension was incubated with a fluorescence βGal probe and subjected to flow cytometry analysis. Representative images of the flow cytometry analysis are shown in (**h**). Percentage of SA-βGal-expressing CD144$^+$ vascular endothelial cells is shown in (**i**). **j–m** The cell suspension prepared as described in (**g**) were subjected to FACS sorting followed by qRT-PCR analysis. mRNA levels of ki67 (**j**), p16 (**k**), p53 (**l**), and p21(**m**) are shown. **n–r** The cell suspension prepared as described in (**g**) were subjected to single-cell imaging flow analysis using ImageStreamX (see the detailed description in the "Methods" section). Representative images of the cells are shown in (**n**). Representative histogram presents max pixel intensity of Ki67 and HMGB1, respectively (**o** and **q**). Quantification of the fluorescence intensity of Ki67 and HMGB1 signals in CD144$^+$ cells were shown in (**p**) and (**r**), respectively. GP growth plate. Ar tissue area. BF bright field. $n = 4$-6 mice. Data are represented as mean ± s.e.m. $*p < 0.05$, $**p < 0.01$, ns not significant, as determined by two-tailed Student's $t$-tests.

p16-iKO as compared with the $p16^{flox/flox}$ (wild-type [WT]) littermates at baseline. Importantly, MPS treatment led to a significant increase in the percentage of SA-βGal-expressing endothelial cells in the metaphysis in WT littermates, but this effect of MPS was blunted in the $p16$-iKO mice (Fig. 3j, k). The type-H vessels (Fig. 3l, m) and OCN$^+$ osteoblasts (Fig. 3n, o) in metaphysis markedly increased in $p16$-iKO mice compared with that in the WT littermates after MPS treatment, suggesting improved angiogenesis and osteogenesis in $p16$-iKO mice. Moreover, MPS treatment-induced decreases in osteoblast number and surface as well as serum P1NP, a marker for bone formation, were largely improved in $p16$-iKO mice vs. WT littermates (Supplementary Fig. 6).

**GC treatment inhibits ANG expression specifically in meta-physis.** To identify the molecular mechanisms by which GCs induce bone vascular endothelial cell senescence, we performed a proteome angiogenesis factor array using bone/bone marrow tissue extracts of femoral metaphysis from vehicle-treated and MPS-treated mice. We reasoned that MPS treatment may induce the reduction of key angiogenesis factor(s) in the bone/bone marrow microenvironment, leading to cell proliferative arrest of the vascular endothelial cells. Four of 53 angiogenesis factors (dipeptidyl peptidase-4, ANG, Collagen XVIII, and aFGF) were dramatically reduced in bone/bone marrow extracts from MPS-treated vs. vehicle-treated mice (Fig. 4a). Of these four factors, ANG is the only secreted factor that can access into endothelial cells and directly regulates cell cycle progression and cell proliferation. To validate our proteome array result, we performed immunofluorescence staining of the bone tissue sections using a specific antibody against ANG. ANG-expressing cells were abundant in primary spongiosa of femoral bones in growing (4-week-old) mice but were much fewer in the same region of adult (16-week-old) mice (Fig. 4b and c). ANG$^+$ cells in primary spongiosa were markedly reduced in MPS-treated mice relative to vehicle-treated mice (upper panel in Fig. 4d and e), and this decrease correlated with MPS-induced reduction in type-H vessels in metaphysis. Scattered ANG$^+$ cells were also found in diaphyseal bone marrow, but MPS treatment did not change the number of these cells (lower panel in Fig. 4d and f), indicating a location-specific ANG expression patterns in bone.

**GC treatment suppresses the formation of ANG-expressing osteoclasts.** Double-immunofluorescence staining was performed to identify the main cell source of ANG. Surprisingly, almost all

the ANG-expressing cells in primary spongiosa adjacent to growth plate are RANK$^+$ (95.37 ± 2.69%) (Fig. 4g and h) and TRAP$^+$ (90.45 ± 4.71%) osteoclasts (Fig. 4j and k). The absolute numbers of RANK$^+$ and TRAP$^+$ cells expressing ANG were significantly reduced in response to MPS treatment (Fig. 4h and k). However, the percentages of ANG-expressing cells in both the RANK$^+$ and TRAP$^+$ cell populations remain the same in MPS-treated mice relative to vehicle-treated mice (Fig. 4i and l), suggesting that MPS primarily induces the diminishment of ANG-expressing osteoclasts. Osteoclasts at the bone/cartilage interface (primary spongiosa) were recently termed a vessel-associated osteoclast subtype that have no bone-resorbing activity but are essential for homeostasis of the type-H vessels in metaphysis[83]. Indeed, four-color immunostaining of the bone tissue sections revealed that ANG-expressing cells were RANK$^+$ osteoclasts that are closely associated with Emcn$^+$ vascular endothelial cells, whereas MPS treatment led to a diminished RANK$^+$ osteoclasts and the ANG expression in the cells in this particular region (Fig. 4m). While MPS treatment induced a reduction in TRAP$^+$ and VPP3$^+$ osteoclasts (Supplementary Fig. 7a–d), the overall macrophage number in bone marrow was not changed in response to MPS treatment (Supplementary Fig. 7e and f). Therefore, the results clearly demonstrate that ANG is exclusively expressed in vascular-associated osteoclasts, which are the primary target cells of GCs (illustrated in Fig. 4n).

In vitro osteoclastogenesis assays were performed to further test the possibility of direct regulation of ANG gene expression by MPS in osteoclasts. The formation of TRAP$^+$ mature multi-nuclear osteoclasts from bone marrow macrophages (BMMs) was suppressed when MPS was added concomitantly with M-CSF and RANKL at the beginning of the culture (Fig. 5ai, bi, and ci). Of note, ANG expression had 4-fold reduction in MPS-treated cells (primarily mononuclear preosteoclasts) relative to vehicle-treated cells (mature osteoclasts) (Fig. 5di). Further, while the number of mature osteoclasts was restored by addition of GR antagonist RU486, ANG expression in these cells was also elevated. The results suggest that mature osteoclasts are a main source of ANG. In another set of experiments, MPS was added at the end stage when multinuclear osteoclasts were already formed (Fig. 5aii). The number of mature osteoclasts was not changed in MPS-treated vs. vehicle-treated cells (Fig. 5bii and cii); and importantly, the expression of ANG was also not affected in response to MPS treatment (Fig. 5dii). These in vitro results, together with the in vivo co-immunostaining data strongly suggest that GCs primarily induce the diminishment of ANG-expressing osteo-clasts rather than modulation of ANG expression via direct gene regulation.

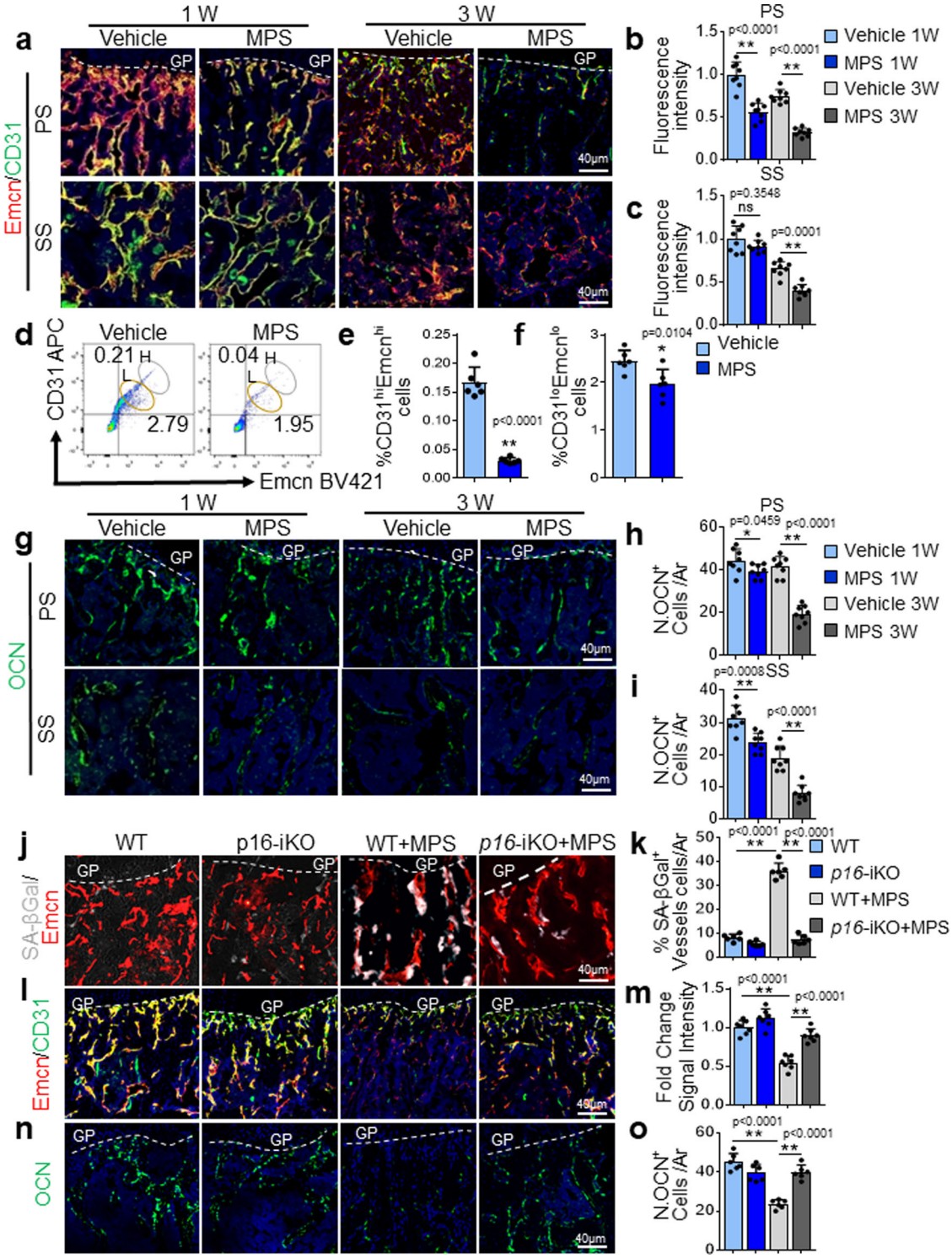

**ANG secreted from metaphyseal osteoclasts mediates osteoclast-vascular crosstalk.** To further determine the role of ANG produced by metaphyseal osteoclasts in regulating vascular endothelial cells, we generated an ex vivo femoral metaphysis explant culture system (Fig. 6a), in which the cultured bone explants were treated with various agents that are known to suppress or activates vascular-associated osteoclasts. We first detected the secretion of ANG from the metaphysis explants and found that the concentration of ANG in the conditioned medium (CM) of the metaphysis culture was much higher than that in serum-free DMEM as detected by ELISA (Fig. 6b, 1st and 2nd

bars). Therefore, ANG protein was indeed secreted by the cells in metaphysis under normal conditions. CM from the metaphysis treated with PTH, a stimulator of vascular-associated osteoclasts, had a higher concentration of ANG relative to control CM (Fig. 5b, 3rd bar). In contrast, CM from the metaphysis treated with alendronate, an inhibitor of vascular-associated osteoclasts, had greatly reduced ANG concentration compared with that in control CM (Fig. 6b, 4th bar), suggesting that ANG is secreted mainly by osteoclasts in metaphysis. Consistent with our in vivo data, ANG concentration in the CM of MPS-treated metaphysis was almost three times lower than the control CM (Fig. 6b, 5th

**Fig. 3 Antagonizing endothelial cell senescence improves GC-impaired bone angiogenesis and osteogenesis. a–i** Three-week-old *BALB/c* mice were treated with MPS at 10 mg/m$^2$/day or vehicle by daily intraperitoneal injection for 1 or 3 weeks. Double-immunofluorescence staining of femur metaphysis sections was performed using antibodies against Emcn (red) and CD31 (green) in (**a**). DAPI stains nuclei blue. Relative yellow fluorescence intensity (vessels expressing both Emcn and CD31) in primary spongiosa (**b**) and secondary spongiosa (**c**) was measured. Cells isolated from femoral metaphysis of MPS-treated or vehicle-treated mice were subjected to flow cytometry analysis. Representative images are shown in (**d**). Black circle: type-H vessels; Brown circle: type-L vascular cells. The percentages of the CD31$^{hi}$Emcn$^{hi}$ cells and CD31$^{lo}$Emcn$^{lo}$ cells are shown in (**e**) and (**f**), respectively. Immunofluorescence staining of femoral metaphysis sections was performed using antibody against Osteocalcin (OCN, green) in (**g**). DAPI stains nuclei blue. Quantified numbers of OCN$^+$ cells per mm$^2$ tissue area (N. OCN$^+$ cells/Ar) in primary spongiosa and secondary spongiosa are shown in (**h**) and (**i**), respectively. **j–o** Three-week-old *Cdh5-Cre$^{ERT2}$;p16$^{flox/flox}$* (*p16*-iKO) mice and *p16$^{flox/flox}$* (WT) mice were injected with tamoxifen (100 mg/kg B.W, 3 doses on the 1st week, 2 doses on the 2nd week, and 1 dose on the 3rd week). At the same date when the 1st dose tamoxifen injection started, mice were also started to receive daily MPS treatment at 10 mg/m$^2$/day or vehicle for 3 weeks. The mice were humanely killed at 6 weeks of age. Representative images of SA-βGal staining (white) and immunostaining of Emcn (red) in primary spongiosa of femoral bone in (**j**). Percentages of SA-βGal-expressing vessels in primary spongiosa were quantified in (**k**). Double-immunofluorescence staining of femur metaphysis sections was performed using antibodies against Emcn (red) and CD31 (green) in (**l**). DAPI stains nuclei blue. Relative yellow fluorescence intensity (vessels expressing both Emcn and CD31) in primary spongiosa was calculated in (**m**). Immunofluorescence staining of femur metaphysis sections was performed using antibody against OCN (green) in (**n**). DAPI stains nuclei blue. Quantified numbers of OCN$^+$ cells per mm$^2$ tissue area (N. OCN$^+$ cells/ Ar) in primary spongiosa are shown in (**o**). GP growth plate, Ar, tissue area. PS primary spongiosa, SS secondary spongiosa. $n = 5$–7 mice. Data are represented as mean ± s.e.m. *$p < 0.05$, **$p < 0.01$, ns not significant, as determined by one-way ANOVA with post hoc Tukey test.

bar). Moreover, co-treatment of the metaphysis explant with RU486, a GC receptor (GR) antagonist, restored the concentration of ANG in the CM to the level of control CM (Fig. 6b, 6th bar), suggesting that the suppressive effect of GCs on ANG production from osteoclasts is GR-dependent.

Because ANG is a potent angiogenesis factor that stimulates endothelial cell growth and proliferation[61,66,67,84], we reasoned that osteoclast-secreted ANG may regulate the activities of the vascular endothelial cells in a non-cell-autonomous manner. To test this hypothesis, we added the CM prepared from the ex vivo bone explant culture to the HUVEC culture. As a positive control, rhANG-stimulated high rate of cell proliferation (Fig. 6c, 1st and 2nd bars) and tube formation (Fig. 6d and e, 1st and 2nd bars) of the HUVECs. We then tested the effects of different CMs on cell proliferation. CM prepared from the metaphysis culture (Control-CM), relative to serum-free control DMEM, stimulated more cell proliferation (Fig. 6c, 3rd bar). The cell proliferation rate was further increased when the cells were incubated with CM from PTH-treated metaphysis (Fig. 6c, 4th bar) but much decreased when the cells were incubated with alendronate-treated metaphysis (AL-CM) (Fig. 6c, 5th bar), or with CM from MPS-treated metaphysis (MPS-CM) (Fig. 6c, 6th bar), relative to Control-CM, consistent with the reduced ANG concentration in the CMs prepared from metaphysis culture with different treatments. Similarly, both rhANG and Control-CM stimulated HUVEC tube formation, while no obvious tube formation was detected in the cells with serum-free DMEM (Fig. 6d and e). Importantly, PTH-CM stimulated more HUVEC tube formation, but AL-CM or MPS-CM stimulated much less HUVEC tube formation than the CM from those treated with CM of the vehicle-treated metaphysis (Fig. 6d and e). These results suggest that ANG from osteoclasts can promote the proliferative capacity and angiogenesis of the neighboring blood vessels.

**The ANG/PLXNB2 axis protects vascular cells from senescence.** To further elucidate the downstream signaling that mediates the effect of ANG on vascular endothelial cells, we investigated the cellular distribution of PLXNB2, which has been identified as a functional ANG receptor[62], in femoral metaphysis. Most Emcn$^+$ vascular endothelial cells (~83.43 ± 4.57%) and a few Osx$^+$ osteoprogenitors (~5.76 ± 3.89%) express PLXNB2 in metaphysis (Fig. 7a), indicating that these cells are ANG-targeting cells. A key downstream event of the ANG/PLXNB2 signaling in endothelial cells is to promote the rRNA transcription[60–62]. The early stage

of rRNA production in cells results in the synthesis of the 47S pre-rRNA, which is cleaved at both ends to generate the 45S pre-rRNA and then processed to produce mature 18S, 5.8S, and 28S rRNAs. We therefore examined whether MPS induces the change of rRNA transcription in vivo by performing simultaneous fluorescence immunostaining and fluorescence in situ hybridization (FISH) of bone tissue sections with an antibody against Emcn and a probe specific to the initiation site of 47S rRNA. Intense positive signal of rRNA transcription was detected in Emcn$^+$ blood vessel cells in metaphysis of vehicle-treated mice, whereas much less positive signal was detected in blood vessels of MPS-treated mice (Fig. 7b and c). Notably, 47S rRNA signal was also detected in non-endothelial cells in metaphysis. However, the signal was not significantly reduced in these cells upon MPS treatment (Fig. 7b and d).

We then tested whether ANG-PLXNB2 signaling is critical for the function of endothelial cells in vitro. As a member of the secreted factor, ANG undergoes receptor-mediated endocytosis from the cell surface to the nucleus to exert its effects on cells[59,62]. We found that knockdown of PLXNB2 in HUVEC cells using siRNA (Fig. 7e) inhibited nuclear translocation of ANG (Fig. 7f) and decreased the activation of the known ANG downstream signaling AKT and ERK phosphorylation (Fig. 7g), validating the requirement of PLXNB2 in mediating the activities of ANG in endothelial cells. The effect of ANG on stimulating endothelial cell proliferation and angiogenesis is mediated through increasing the rRNA transcription in the nucleus[60–62]. We assessed whether knockdown of PLXNB2 affects the RNA synthesis by performing the Click-iT RNA-imaging assay, in which newly synthesized RNA can be detected in the cells fed an alkyne-modified nucleoside 5-ethynyl uridine (EU). Cells transfected with scrambled control siRNA showed an intense EU-labeled RNA signal inside nucleoli after 2 h of exposure to EU, and the signal increased after 6 h of exposure (Fig. 7h, 1st row). Treatment of the rhANG significantly increased the intensity of EU-labeled RNA signal inside nucleoli (Fig. 7h, 2nd row). In the cells transfected with PLXNB2 siRNA, the neo-transcribed RNA was detected as indicated by a strong decrease in EU-labeled RNA signal inside nucleoli after both 2 and 6 h of exposure to EU in both rhANG untreated (Fig. 7h, 3rd row) and treated groups (Fig. 7h, 4th row). These results suggest that PLXNB2 is required for ANG-stimulated rRNA synthesis in endothelial cells. Because the defects of rRNA transcription and ribosome biogenesis contribute to cellular senescence[53], we examined whether knockdown of PLXNB2 leads to endothelial cell senescence. Increased number

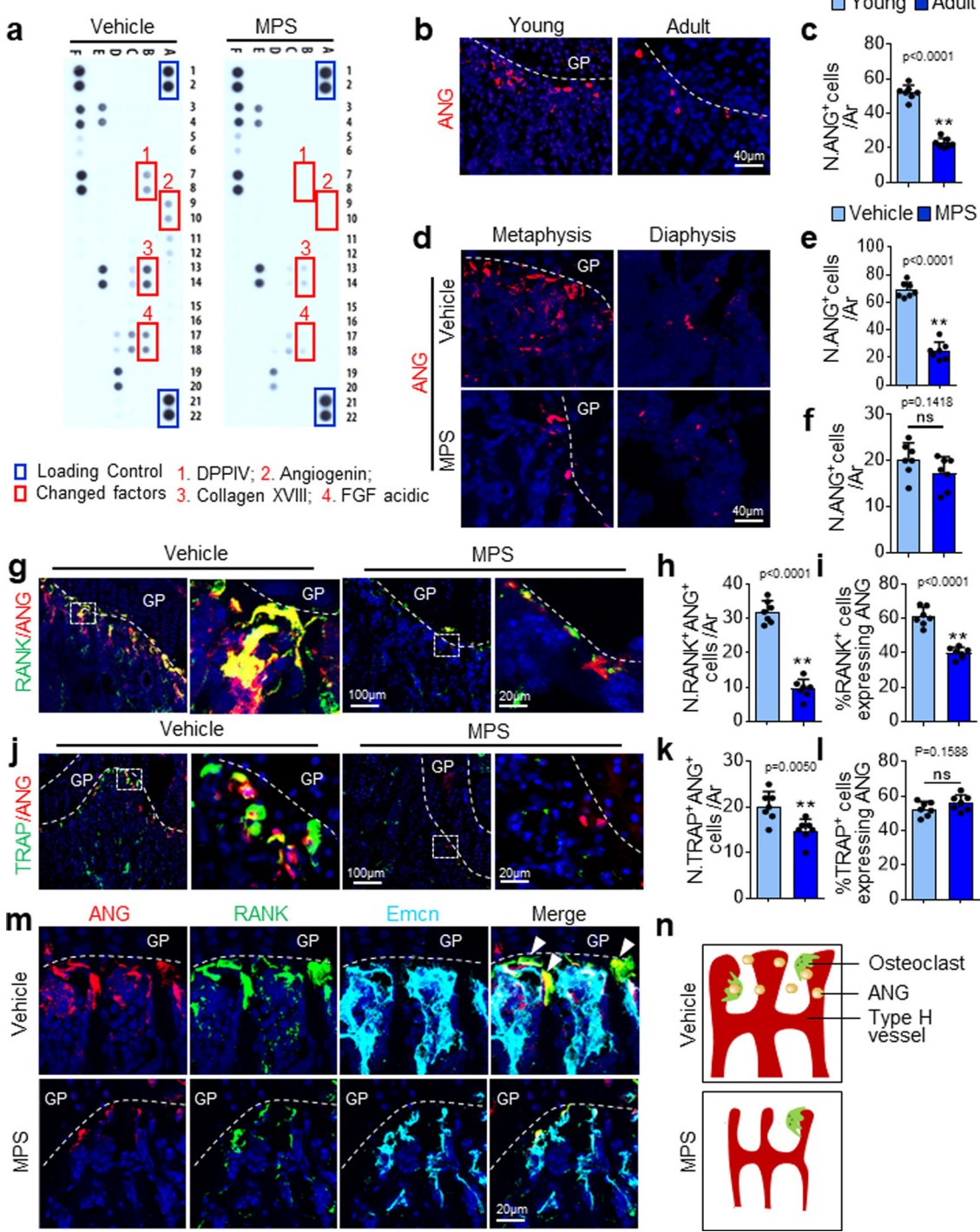

of SA-βGal⁺ Cells was detected in the cells transfected with PLXNB2 siRNA relative to the cells transfected with control siRNA (Fig. 7i). Lamin B1 loss is a senescence-associated biomarker[85]. Much less lamin B1 expression was detected in PLXNB2 siRNA-transfected cells than in control siRNA-transfected cells (Fig. 7j, 1st panel). Moreover, cell proliferation marker Ki67 expression was reduced (Fig. 7j, 2nd panel) but cell senescence marker p21 expression was increased (Fig. 7j, 3rd panel) in the cells transfected with PLXNB2 siRNA relative to the cells transfected with control siRNA. Loss of ANG signaling was also shown to induce apoptosis in other cell types[86–88]. However, increased cell apoptosis was not detected in HUVECs transfected with PLXNB2 siRNA relative to the cells transfected with control

siRNA as detected by TUNEL assays (Supplementary Fig. 8a). We also took advantage of an ANG neutralizing antibody (26-2F) and an antibody specifically blocking the ANG-PLXNB2 binding (mAB17)[62] and conducted tube formation assays. Intriguingly, both antibodies completely abolished angiogenic activity induced by the CM prepared from metaphysis explant culture (Fig. 7k and l), further demonstrating the function of the ANG/PLXNB2 axis as a key signaling in promoting angiogenic activity of endothelial cells.

**rhANG rescues GC-impaired bone growth and mineral acquisition.** We examined whether systemic administration of

**Fig. 4 GC treatment induces a reduction in ANG expression in vascular-associated osteoclasts. a** Three-week-old *BALB/c* mice were treated with MPS at 10 mg/m$^2$/day or vehicle by daily intraperitoneal injection for 2 weeks. Bone tissue extracts of femoral metaphysis were prepared. Relative levels of 53 mouse angiogenesis-related proteins in the tissue extracts were simultaneously assessed using the Proteome Profiler Mouse Angiogenesis Array Kit. The factors differentially expressed were marked with red squares. **b** and **c** Immunofluorescence staining of femoral metaphysis sections of four-week-old (Young) and four-month-old (Adult) *C57BL/6J* mice were performed using antibody against ANG. Representative images are shown in (**b**). Red: ANG; Blue: DAPI. Quantified numbers of ANG$^+$ cells in primary spongiosa per mm$^2$ tissue area (N. ANG$^+$ cells/Ar) are shown in (**c**). **d–k** Three-week-old *C57BL/6J* mice were treated with MPS at 10 mg/m$^2$/day or vehicle by daily intraperitoneal injection for 2 weeks. Immunofluorescence staining of femur metaphysis sections were performed using antibodies against ANG. Representative images are shown in (**d**). Red: ANG; Blue: DAPI. Quantified numbers of ANG$^+$ cells in primary spongiosa and diaphyseal bone marrow per mm$^2$ tissue area (N. ANG$^+$ cells/Ar) are shown in (**e**) and (**f**), respectively. Double immunofluorescence staining of femoral metaphysis sections was performed using antibodies against ANG and RANK or ANG and TRAP. Representative images are shown in (**g**) and (**j**), respectively. Red: ANG; Green: RANK or TRAP; Blue: DAPI. Boxed areas are shown at a higher magnification in corresponding panels to the right. Quantified numbers of RANK-double and ANG-double positive cells (N. RANK$^+$ANG$^+$ cells/Ar) and TRAP- and ANG-double positive cells (N. TRAP$^+$ANG$^+$ cells/Ar) are shown in (**h**) and (**k**), respectively. Percentages of ANG-expressing cells in the RANK$^+$ and TRAP$^+$ cell populations (%RANK$^+$ cells expressing ANG and %TRAP$^+$ cells expressing ANG) are shown in (**i**) and (**l**), respectively. **m** Four-color immunofluorescence staining of femur metaphysis sections was performed using antibodies against ANG (red), RANK (green), and Emcn (cyan). DAPI stains nuclei. Note: cells in yellow (ANG-expressing osteoclasts) are closely associated with the cells in cyan (Emcn$^+$ vascular endothelial cells). Arrows indicate RANK$^+$ANG$^+$ double-positive cells reside closely to Emcn$^+$ vascular endothelial cells. **n** Schematic diagram showing ANG-expressing osteoclasts that reside closely to type-H vessels in primary spongiosa in the absence or presence of MPS treatment. GP growth plate. $n = 4$–8 mice. Data are represented as mean ± s.e.m. **$p < 0.01$, ns not significant as determined by two-tailed Student's $t$-tests.

rhANG rejuvenates senescent bone blood vessels in MPS-treated young mice. While MPS treatment induced accumulated vascular endothelial cell senescence, as indicated by increased SA-βGal expression, the senescent endothelial cells were reduced by rhANG co-treatment (Fig. 8a and b). We then assessed the changes of angiogenesis and osteogenesis. rhANG alone did not change the number of type-H vessels in metaphysis of young mice. However, MPS-induced impairment of type-H vessels were markedly improved by rhANG co-treatment (Fig. 8c and d). Similarly, MPS-induced declines in osteogenesis were also largely rectified by rhANG co-treatment, as detected by immunofluorescence staining of Osx of the bone tissue sections (Fig. 8e and f). MPS treatment reduced osteoblast number and surface (Supplementary Fig. 9a–c) as well as the number of osteocytes (Supplementary Fig. 9d and e), which were all largely rectified by rhANG co-treatment. The osteoclast number and surface were not changed by rhANG treatment (Supplementary Fig. 9f–h). Finally, we evaluated whether rhANG prevents GC-impaired bone growth and bone mass acquisition. MPS-treated mice had reductions in tail length and body weight indicating impaired growth (Supplementary Fig. 10a and b). Intriguingly, rhANG co-treatment significantly increased tail length in male mice (Supplementary Fig. 10a). The changes in body weight were not significant in the combined rhANG and MPS treatment group relative to the MPS treatment group (Supplementary Fig. 10b). H&E staining of the tissue sections showed shorter height of the distal femur growth plate in MPS-treated mice (vs. vehicle-treated mice), and rhANG largely reversed this negative effect (Supplementary Fig. 9c). The heights of the proliferative zone and hypertrophy zone were shorter in MPS-treated mice, but the height of the resting zone was unaffected (Supplementary Fig. 10c–f). Importantly, rhANG co-treatment improved the heights of the hypertrophy zone and proliferative zone, which were impaired by MPS (Supplementary Fig. 10c–e). μCT analysis showed that combined rhANG and MPS treatment, relative to MPS treatment alone, allowed significantly longer femur growth (Supplementary Fig. 10g). Moreover, the MPS-induced low-bone-mass phenotype in the metaphysis of the femur (Fig. 8g–l), characterized by low trabecular BV/TV (Fig. 8i), Tb. Th (Fig. 8j), and Tb. N (Fig. 8k), as well as an increase in Tb. Sp (Fig. 8l), was partially rescued by rhANG co-treatment. Of note, rhANG did not change any of these parameters in the vehicle-treated mice. Improved bone formation by rhANG was also confirmed by Goldner's Trichrome staining and double calcein labeling

analysis. Femoral metaphysis of MPS-treated mice showed a decrease in newly formed bone (red), which was improved by rhANG co-treatment (Fig. 8m and n). Double calcein labeling showed that the distance between two consecutive labels was less in MPS-treated mice than in the vehicle-treated control mice and became more in rhANG-cotreated mice (Fig. 8o). The reduced BFR/BS in response to MPS treatment was improved by rhANG co-treatment (Fig. 8p). Therefore, rhANG successfully protects growing bone from the negative effects of GC treatment on longitudinal bone growth and bone formation.

## Discussion

Although the negative regulation of bone angiogenesis by GCs in young animals was previously noted[35,36], the mechanisms mediating this effect of GCs are largely unknown. Here, we provide evidence that ANG secreted from metaphyseal osteoclasts is essential to maintain the closely associated blood vessels in growing long bone through an ANG/PLXNB2-rRNA transcription signaling (Fig. 9a). We further reveal that GC treatment inhibits ANG production through suppression of osteoclast formation in metaphysis, leading to blood vessel cell senescence and impairment of angiogenesis with coupled osteogenesis (Fig. 9b). Thus, our work uncovers a new line of cellular and molecular mechanisms for the deleterious effects of GCs on the growing skeleton.

Previously, we reported that in healthy long bone, cells in the primary spongiosa zone of the metaphysis are highly proliferative during early puberty but undergo programmed cellular senescence during late puberty, when the speed of bone growth/accrual starts to slow[71]. In the present study, we found that 3-week-old (prepubertal to early pubertal) mice treated with MPS acquire an earlier and exacerbated cellular senescence phenotype in the same region of growing long bone. We provide a comprehensive measure of senescence in MPS-treated mice. Using the knock-in *p16$^{tdTom}$* reporter mice, we observed significantly increased tdTom$^+$ cells in metaphysis of long bone as early as 1 week and peaking at 2 weeks after MPS treatment, as detected by flow cytometry and in situ fluorescence analysis. The cellular senescence induced by MPS was also confirmed by other markers, including SA-βGal, loss of expression of Ki67, and relocalization of HMGB1 from the nucleus to the cytoplasm. Childhood is a critical period for optimizing bone growth and mineral accrual. Bone mass and strength during this period are influenced by

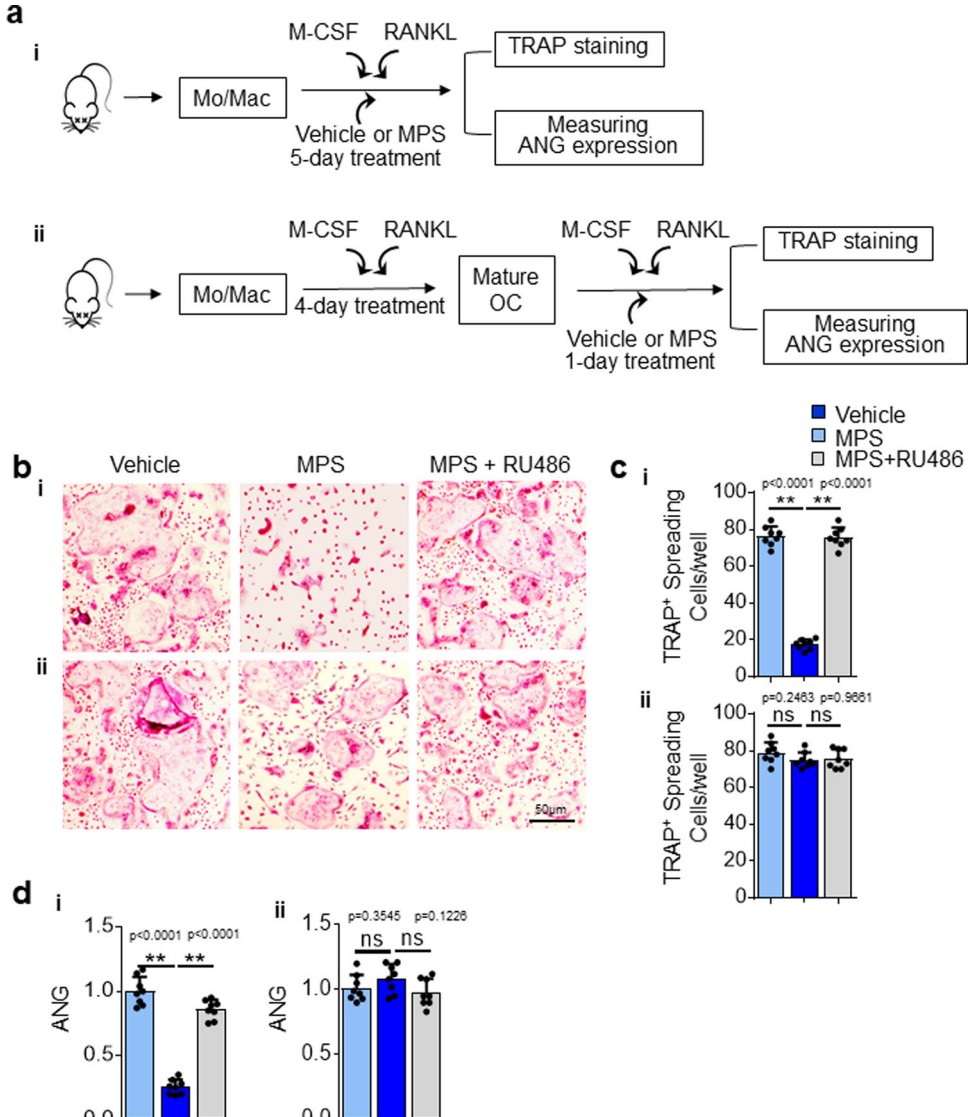

**Fig. 5 GC Treatment suppresses osteoclast formation but does not directly regulate ANG gene expression in osteoclasts. a** Schematic diagram showing the procedure of the in vitro experiments. Bone marrow mononuclear cells/macrophages were isolated from 3-week-old *C57BL/6J* mice and cultured with M-CSF and RANKL for 5 days to induce the formation of mature osteoclasts. The cells were treated with vehicle or MPS with or without RU486 for the entire 5 days (i) or only the last day (ii). The cells were subjected to TRAP staining or lysed to measure the mRNA expression level of ANG. TRAP staining of the cells is shown in (**b**). Quantified number of TRAP+ spreading multinucleated cells/well is shown in (**c**). The levels of ANG expression were analyzed by qRT-PCR (**d**). Experiments were performed in triplicate and were repeated three times. Data are represented as mean ± s.e.m. **\*\*p < 0.01, ns, not significant as determined by one-way ANOVA with post hoc Tukey test.

genetic and epigenetic factors, activity, nutrition, and hormones[13,89,90]. The metaphysis of long bone, which contains primary and secondary spongiosa, is a specialized region in which cellular components change substantially, and the cells proliferate actively during childhood bone growth, both in size and mass. Our data suggest that cells in this region are more vulnerable than the cells in the cortical-rich diaphysis, making it easier for them to acquire the senescence phenotype in response to hormonal changes/external stimuli such as GCs.

Exogenous GCs exert multiple effects on skeleton and regulate many cell types[18–21,26–28]. GCs prolong the lifespan of osteoclasts, increase apoptosis of osteoblasts and osteocytes, and reduce the number of osteoprogenitors from MSCs by promoting the adipogenic differentiation pathways. Our results show that bone vascular cells are another cell target of GCs. We found that vascular endothelial cells in type-H vessels, which are highly proliferative and osteogenesis-coupled vessels in the metaphysis,

undergo GC-induced senescence in young mice. As a result, blood vasculature and osteogenesis diminish in this region. In growing long bone, angiogenesis in the metaphyseal region is a prerequisite for high rates of longitudinal growth and mineral acquisition[91] and therefore is essential for skeletal health in children. We further demonstrate that selective deletion of *p16INK4a* in vascular endothelial cells greatly improved angiogenesis with coupled osteogenesis that was impaired by GC treatment and rescued the growth retardation and bone loss phenotype. Therefore, bone blood vessel cell senescence plays a causal role in GC-induced negative effects on growing skeleton.

We identify a mechanism through which type-H vessels in the metaphysis of growing bone are maintained by a specific subtype of osteoclasts. We show that in the metaphyseal microenvironment, ANG secreted from osteoclasts acts on nearby vascular endothelial cells via binding to its receptor PLXNB2, through which ANG is endocytosed into the endothelial cells and

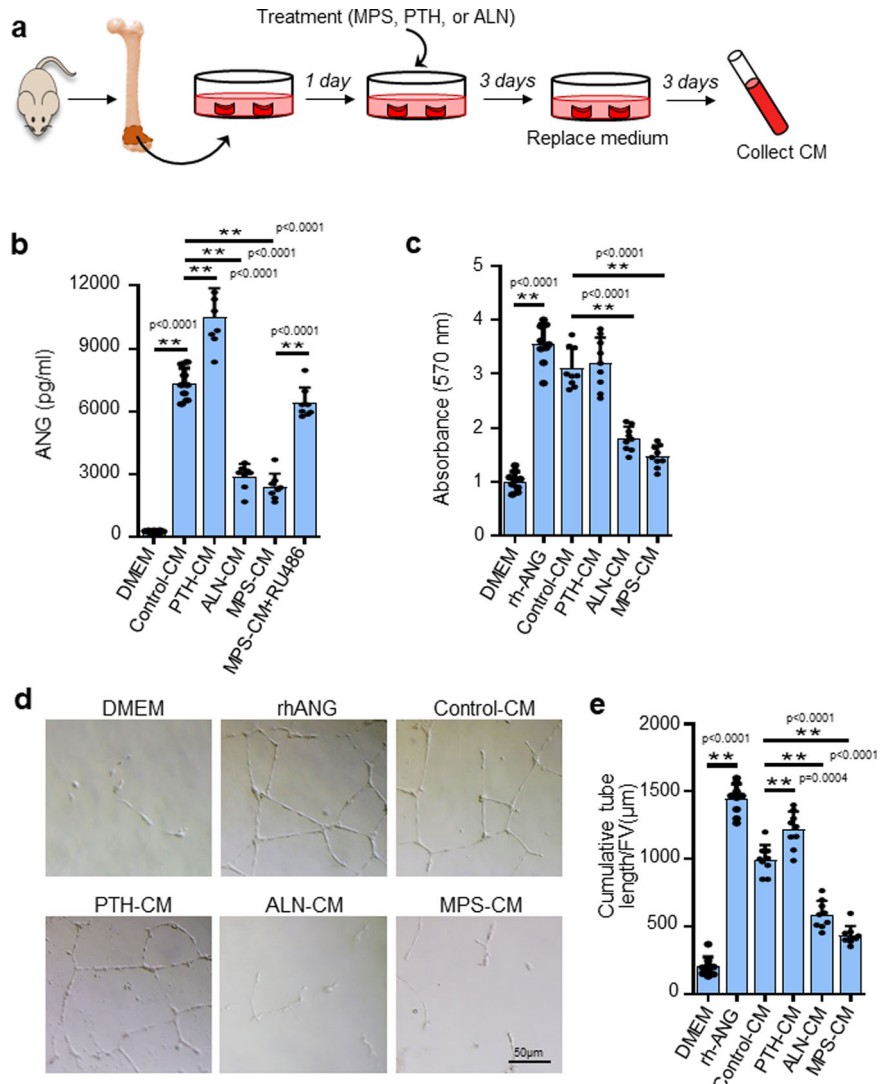

**Fig. 6 ANG secreted from metaphyseal osteoclasts mediates osteoclast-vascular crosstalk. a** Schematic diagram showing the procedure of the ex vivo experiments. Metaphyseal bone of distal femur was isolated from 3-week-old *C57BL/6J* mice and cultured in serum-free DMEM for 1 day to adjust to the ex vivo environment. The explant was then cultured for 3 days with different treatments. Fresh medium was replaced after washing away the medium. Conditioned medium (CM) was collected after another 3-day culture. **b** The concentrations of ANG in the CMs were measured by ELISA. **c** Different CMs were individually added in the HUVEC culture, and the proliferation of the cells was measured by MTT assay. **d** and **e** Different CMs were individually added in the HUVEC culture planted on Matrigel. Tube formation assay images are shown in (**d**). Quantitative analysis of cumulative tubule length is shown in (**e**). PTH parathyroid hormone, ALN alendronate. Experiments were performed in triplicate and were repeated three times. Data are represented as mean ± s.e.m. **$p < 0.01$ as determined by one-way ANOVA with post hoc Tukey test.

translocated into the nucleus for rRNA transcription, resulting in cell proliferation and eventual blood vessel growth. Therefore, ANG functions as a cell non-autonomous manner in this process. In a previous report, Romeo et al.[53] identified a distinct subset of osteoclasts that are closely associated with type-H capillaries in the metaphyseal region. This subtype of osteoclasts functions to regulate the growth of type-H vessels. Our results agree with their finding and provide a molecular mechanism for the osteoclast–vascular cell communication in this specific skeletal region. We show that ANG/PLXNB2 as a key signaling mediator of this process to maintain angiogenesis during skeletal growth. Previous studies from Hu's laboratory show that ANG is expressed in bone marrow osteolineage cells to regulate the function of hematopoietic stem/progenitor cells (HSPCs)[65,92]. We found that ANG⁺ cells are almost exclusively expressed in RANK⁺TRAP⁺ osteoclasts in primary spongiosa under the growth plate. Scattered ANG⁺ cells were also detected in bone marrow of the central part

of diaphyseal bone (Fig. 4d). The cell lineages that express ANG in the diaphyseal bone marrow in our model remain to be determined. Nevertheless, our results and the findings of Hu's group suggest that the cell source of ANG and its function are apparently distinct in different regions of bone and may also be growth phase-dependent.

An important question is that how GCs inhibit ANG production from metaphyseal osteoclasts. Our in vivo co-immunofluorescence-staining data show that the percentages of ANG-expressing cells in osteoclasts remain the same although there is a significant reduction in the absolute numbers of osteoclasts expressing ANG in metaphysis. Therefore, it is likely that GCs induce the diminishment of ANG-expressing osteoclasts rather than modulation of ANG gene expression directly. Our results from the ex vivo metaphysis explant culture experiment, showing that osteoclast stimulator PTH increased the ANG production and osteoclast inhibitor alendronate decreased ANG

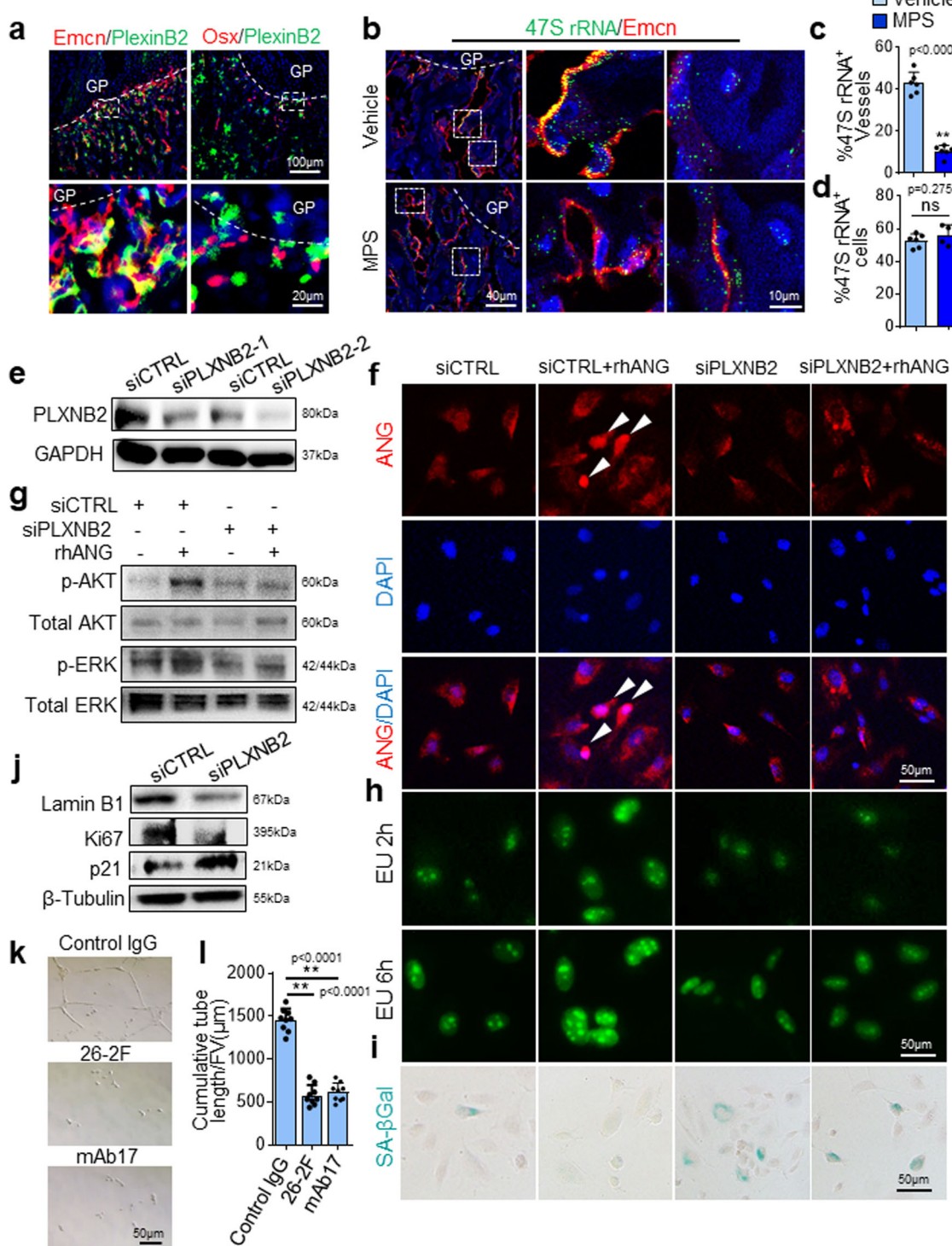

production, support the above assumption. More importantly, the finding from our in vitro osteoclastogenesis study further validates that ANG is derived from mature osteoclasts, and the inhibitory effect of GCs on ANG production in metaphysis is primarily through suppressing osteoclast formation. It has long been recognized that exogenous GCs have divergent effects on osteoclasts both in vivo and in vitro due to differences in strain, age, glucocorticoid dose, or experimental set-up[93]. Our finding that MPS treatment decreases osteoclast numbers in femoral metaphysis in growing young mice is consistent with previous work conducted using the same age of young mice[74,75].

Moreover, our in vitro result that MPS treatment suppresses mature osteoclast formation also agrees with the earlier finding that higher doses of GCs suppress the proliferation of osteoclast precursors and interfere with cytoskeletal reorganization[18]. GC treatment has been shown to result in increased osteoclasts in adult skeleton[94,95]. The different effects on osteoclasts exerted by GCs are likely attributable to the differences in the age of the mice, the bone region for analysis, and the dosage of GCs.

Given that osteoprogenitor cells are abundant in metaphysis and highly proliferative in growing skeleton, we postulate that this cell lineage, in addition to vascular cells, may be another

**Fig. 7 The ANG/PLXNB2 axis is essential to maintain the proliferative activity of vascular endothelial cells and protect them from senescence. a–d**
Three-week-old *C57BL/6J* mice were treated with MPS at 10 mg/m$^2$/day or vehicle by daily intraperitoneal injection for 2 weeks. Double immunofluorescence staining of femoral sections using antibodies against PLXNB2 (green) and Emcn (red) or Osx (red) (**a**). Simultaneous fluorescence immunostaining of Emcn (red) and FISH analysis of 47S rRNA transcription (green) in bone tissue sections were shown in (**b**). DAPI stains nuclei blue. Boxed areas are shown at a higher magnification in corresponding panels to the right. Quantitative analysis of percentage of Emcn$^+$ blood vessels in (**c**) and non-blood vessel cells (**d**) that show positive signal of 47S rRNA transcription in primary spongiosa. Blood vessels with more than 20 green dots are considered as positive. Non-blood vessel cell with more than five green dots in nucleus are considered positive. **e** Immunoblot of PLXNB2 in HUVECs transfected with two individual PLXNB2 siRNAs (siPLXNB2-1, -2) or scrambled control siRNA (siCTRL). **f–i** HUVECs were transfected with siCTRL or siPLXNB2 for 36 h and then stimulated with 200 ng/mL rhANG or vehicle for another 12 h. Cells were fixed and immunofluorescence staining was performed using antibody against ANG (red) in (**f**). Arrow heads indicate nuclear translocation of ANG. DAPI stains nuclei blue. Immunoblot was performed using antibodies against p-AKT, AKT, p-ERK, or ERK in (**g**). Click-iT RNA-imaging assays were performed, and confocal images of the cells pulsed with EU for 2 and 6 h were obtained (green: Alexa 488-EU) in (**h**). Cells were subjected to SA-βGal staining with representative images in (**i**). **j** HUVECs were treated transfected with siCTRL or siPLXNB2 for 48 h. Immunoblot was performed using individual antibodies against LaminB1, Ki67, p21, and GAPDH. **k, l** HUVEC was seeded on Matrigel with control IgG, 26-2F or mAb17. Images of tubule formation is shown in (**k**). Quantitative analysis of cumulative tubule length is shown in (**l**). **p < 0.01 as determined by one-way ANOVA with post hoc Tukey test. GP growth plate. *n* = 4-6 mice, Data are represented as mean ± s.e.m. **p < 0.01, ns not significant as determined by two-tailed Student's *t*-tests.

target of ANG. Our data show that most vascular endothelial cells have PLXNB2 expression, but only a small portion of Osx$^+$ osteoprogenitor cells are PLXNB2-positive, suggesting that osteoprogenitors in metaphysis may not be a main direct target of ANG. However, in addition to the senescent vascular endothelial cells, we did detect increased number of senescent Osx$^+$ osteo-progenitor cells in response to GC treatment. One possible explanation for the discrepancy is that the senescent endothelial cells, directly caused by ANG/PLXNB2 downregulation, may secrete SASP factors, which further induce the senescence of the neighboring osteoprogenitor cells in an effect known as "bystander" senescence[96–98]. Given that angiogenesis is always coupled with osteogenesis and that the vascular endothelial cells and osteoprogenitors are in close proximity in this region[73], our assumption is not implausible. Our data show that GC treatment also stimulates cell apoptosis in the metaphysis of long bone, mainly in the osteoprogenitor population. These results suggest that, in response to GC treatment, vascular endothelial cells primarily undergo senescence, whereas osteoblast lineage may have multiple cell fate changes, such as cell apoptosis, cell senescence, and lineage shift[27,28,99,100]. GC treatment induces autophagy in osteocyte[101], which is required for osteocyte death[102]. It has also been shown that a lower dose of GCs acti-vates autophagy, and a higher dose increases apoptosis[103]. It will be informative to test whether GC treatment stimulates autop-hagy in vascular endothelial cells in metaphysis and how autop-hagy plays a role in the senescence of this cell lineage in response to GCs.

Both our in vivo and in vitro work suggests that loss or downregulation of the ANG/PLXNB2 axis contributes to GC-induced cellular senescence. Knockdown of PLXNB2 in endo-thelial cells was sufficient to inhibit ANG nuclear translocation and induce cell senescence in the absence of MPS treatment. Given the facts that ANG is an important factor for stimulating ribosome biogenesis[68,69,104] and that the reduced ribosome bio-genesis contributes to cellular senescence[53], it is reasonable to postulate that a deficit in the ANG/PLXNB2-ribosome biogenesis pathway is not only involved in bone blood vessel senescence induced by GCs but also an important common mechanism to initiate vascular endothelial cell senescence. The mechanisms by which the downregulation of ANG/PLXNB2 axis leads to an rRNA transcription deficit remain to be determined. It was reported that ANG inhibits H3K9 methylation and activates H3K4 methylation at the ribosomal DNA promoter, suggesting an involvement of epigenetic regulation[58]. Interestingly, we pre-viously found that Ezh2-H3K27me3 loss is a key mechanism for the occurrence of cellular senescence in the metaphyseal region of

growing bone[71]. Future investigation of the involvement of Ezh2-H3K27me3 in endothelial cellular senescence induced by ANG/PLXNB2 downregulation is important.

Finally, we demonstrate that systemic administration of rhANG in mice rescued GC-impaired bone growth and bone loss in long bone of young mice. Further testing the beneficial effect of rhANG administration on the changes of the skeleton at other locations is needed to determine its translational potential. GC treatment has been shown to induce bone loss in adult lumbar vertebrate[94,95,105]. It is interesting to investigate whether GCs also leads to vascular cell senescence in vertebrae and whether rhANG administration improves GC-impaired bone mass and quality at this location. Nevertheless, our finding opens new possibilities for the treatment of GC-bone complications in the pediatric popu-lation. Systemic administration of rhANG in mice with amyo-trophic lateral sclerosis in preclinical studies improved vascular network maintenance and motoneuron survival[65,106], represent-ing a promising therapeutic strategy. It is reasonable that future studies consider agents/approaches that activate the ANG−ribo-some biogenesis pathway as a therapeutic schema for pediatric osteoporosis or osteonecrosis, especially for patients receiving long-term GC treatment.

## Methods

**Animals and treatment**. We purchased the *Cdh5-Cre^{ERT2}* mice (*C57BL/6* back-ground, stock no. 13073) from Taconic Biosciences (Derwood, MD). The *BALB/c* (stock no. 000651) and *C57BL/6J* (stock no. 000664) mice were purchased from the Jackson Laboratory (Bar Harbor, ME). *p16^{flox/flox}* mice were generated by Dr. Gloria H. Su's laboratory from the Department of Pathology, Columbia University Medical Center[107]. *p16^{tdTom}* reporter mice (*C57BL/6* background) were generated by Dr. Norman E. Sharpless's laboratory from the Curriculum in Genetics and Molecular Biology, University of North Carolina School of Medicine (Chapel Hill, NC)[72].

We crossed the *Cdh5-Cre^{ERT2}* mice with *p16^{flox/flox}* mice (*C57BL/6* background). The offspring were intercrossed to generate *Cdh5-Cre^{ERT2}; p16^{flox/flox}* (*p16* iKO) and *p16^{flox/flox}* (*WT*) mice. To induce *CreER* activity, we injected mice at designed time points with tamoxifen (100 mg/kg.B.W.). To avoid the effect of Tamoxifen on the outcome measures, both *p16^{flox/flox}* mice (*WT* littermates) and *Cdh5-Cre^{ERT2}; p16^{flox/flox}* mice were treated with tamoxifen in all the treatment groups. The genotypes of the mice were determined by PCR analyses of genomic DNA extracted from mouse-tail snips using the following primers: *Cdh5-Cre^{ERT2}* forward, 5′-GCG GTC TGG CAG TAA AAA CTA TC-3′ and reverse, 5′-GTG A AA CAG CAT TGC TGT CAC TT-3′; *loxP p16* allele forward, 5′-AGG AGT CCT GGC CCT AGA AA-3′ and reverse, 5′-CCA AAG GCA AAC TTC TCA GC-3′.

To detect the GC-induced bone vascular phenotype, we injected 3-week-old *BALB/c* mice and 3-week-old *C57BL/6J* mice with MPS (10 mg/m$^2$/day) for 1 or 3 weeks. As we obtained similar results regarding the type H vessel change and metaphyseal cellular senescence from these two different species, only the results from *BALB/c* mice are presented in this study. Body surface area (BSA) was calculated using Meeh's formula, with a *k* constant of 9.82 for mice × body weight (g) to the two-thirds power (BSA = kW$^{2/3}$). For ANG administration, 3-week-old *BALB/c* mice were treated with rhANG (1 μg/day, R&D systems, 265-AN) as

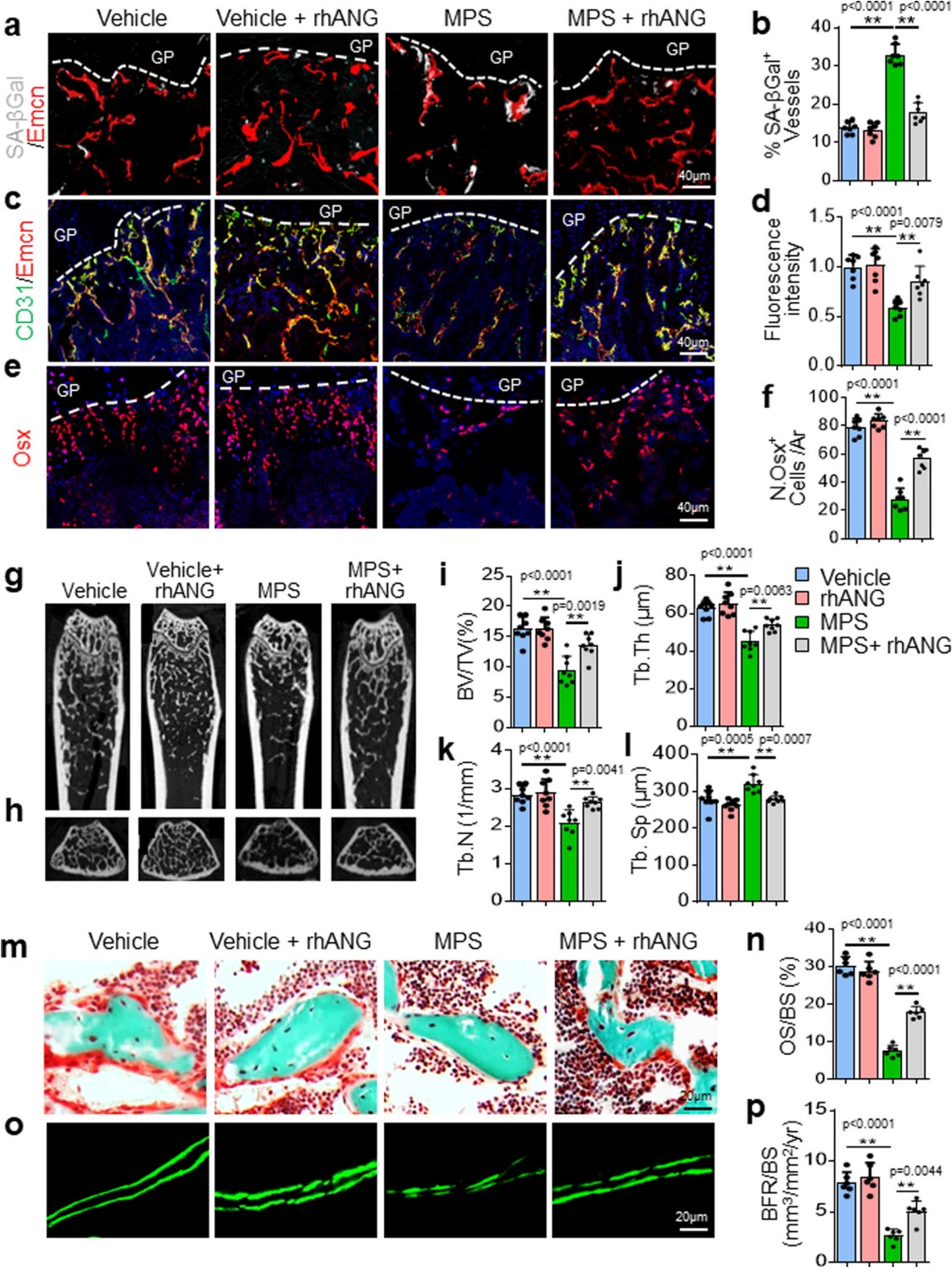

previously described[108]. In some experiments, combined treatment of rhANG (1 µg/day) and MPS (10 mg/m²/day) was used.

Mice were housed in a normal condition with 12:12 h light:dark cycle in a temperature-controlled room with food and water ad libitum. All animals were maintained in the animal facility of The Johns Hopkins University School of Medicine under protocol MO18M139 approved by the Institutional Animal Care and Use Committee of The Johns Hopkins University, Baltimore, MD. All protocols and procedures followed the guidelines of the Institutional Animal Care and Use Committee of The Johns Hopkins University, Baltimore, MD.

**SA-βGal staining and immunofluorescence staining of bone tissue sections**. Mice femora were dissected after sacrifice, and adherent muscles were removed.

The bones were fixed overnight in 10% formalin at 4 °C. After washing with PBS, the femur was decalcified in 0.5 M EDTA (pH 7.4) at 4 °C with constant shaking for 3–4 days and then dehydrated in 20% sucrose and 2% polyvinylpyrrolidone solution for 24 h. Finally, the tissues were embedded in OCT, and 20-µm-thick, longitudinally oriented bone sections were collected for staining. Senescent cells were detected using a senescence βGal staining kit according to the manufacturer's instructions (Cell Signaling Technology, Danvers, MA). For immunofluorescence staining, we incubated the sections with primary antibodies to Emcn (Santa-Cruz, sc-65495, 1:50), CD31 (R&D Systems, Inc, Minneapolis, MN, FAB3628G, 1:100), ANG (rabbit monoclonal, C527, 1:200, generated by the laboratory of Dr. Guo-Fu Hu), Plexin-B2 (1:200, eBioscience, eBio3E7), Osx (Abcam, ab22552, 1:200), OCN (Takara, M188, 1:200), F4/80 (Abcam, ab6640, 1:200), RFP (Rockland antibodies & assays, 600-401-379, 1:200) and HMGB1 (Abcam, ab18256, 1:300) overnight at 4 °

**Fig. 8 Recombinant human ANG (rhANG) rescues GC-impaired bone growth and mineral acquisition.** Three-week-old *BALB/c* mice were treated with vehicle, MPS alone at 10 mg/m$^2$/day or MPS plus rhANG (1 μg/day) by daily intraperitoneal injection for 4 weeks. Representative images of SA-βGal staining (white) and immunofluorescence staining of Endomucin (Emcn, red) in primary spongiosa of femoral bone in (**a**). Percentage of SA-βGal-expressing vessels were quantified in (**b**). Double immunofluorescence staining of Emcn (red) and CD31 (green) in metaphysis of femoral bone in (**c**). DAPI-stained nuclei blue. Relative yellow fluorescence intensity in primary spongiosa was measured in (**d**). Immunofluorescence staining of femoral metaphysis sections were performed using antibody against Osx (red) in (**e**). DAPI stains nuclei blue. Numbers of Osx$^+$ cells per mm$^2$ tissue area (N. Osx$^+$ cells/Ar) were quantified in (**f**). Representative μCT images of distal femur in mice were shown in (**g**, longitudinal sections) and (**h**, cross sections). Quantitative analyses of trabecular bone volume fraction (BV/TV) (**i**), trabecular thickness (Tb. Th) (**j**), trabecular number (Tb. N) (**k**), and trabecular separation (Tb.Sp) (**l**). Trichrome staining of the metaphyseal trabecular bone at distal femora in (**m**). Osteoid stains red and mineralized bone stains green. Osteoid surface per bone surface (OS/BS) was measured in (**n**). Representative images of calcein double labeling (**o**) and quantification of bone formation rate per bone surface (BFR/BS) (**p**) of the metaphyseal trabecular bone at distal femora. GP growth plate. $n = 6$-10 mice. Data are represented as mean ± s.e.m. \*\*$p < 0.01$ as determined by one-way ANOVA with post hoc Tukey test.

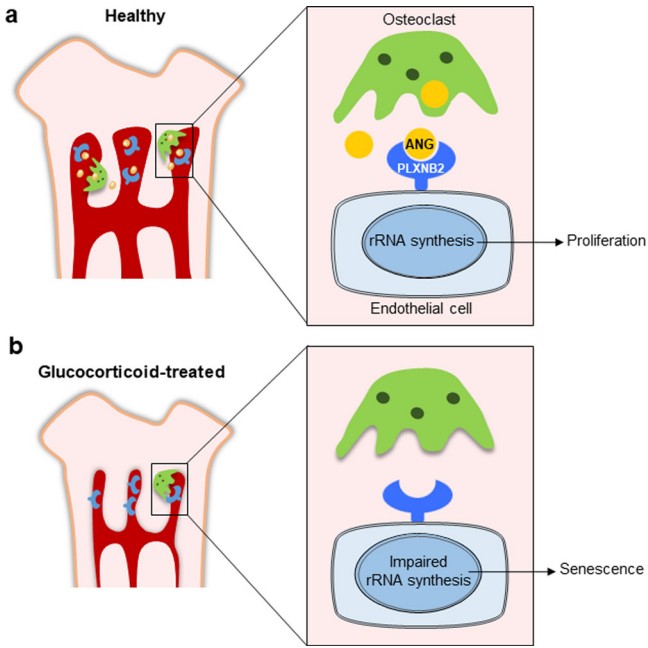

**Fig. 9 Schematic model for the ANG/PLXNB2-rRNA transcription signaling in mediating osteoclast-endothelial cell crosstalk in metaphysis of growing bone. a** In normal growing mice, ANG secreted from metaphyseal osteoclasts maintains the proliferation of neighboring blood vessel endothelial cells in long bone through ANG/PLXNB2-rRNA transcription signaling. **b** GC treatment inhibits ANG production through suppression of osteoclast formation in metaphysis, leading to blood vessel cell senescence and impairment of angiogenesis with coupled osteogenesis.

C, followed by incubation with FITC, or Cy3-conjugated secondary antibodies (Jackson ImmunoResearch, 1:200). Nuclei were counterstained with DAPI (Sigma). The sections were mounted with the ProLong Antifade Kit (Molecular Probes) and observed under a Zeiss LSM780 confocal microscope. For simultaneous SA-βGal staining and immunofluorescence staining, 10 μm tissue sections were first subjected to a βGal staining using the senescence βGal staining kit according to manufacturer's instructions (Cell Signaling Technology) followed by immunofluorescence staining using the same method described above. To obtain the co-localization image, an immunofluorescence imaging was obtained first followed by a brightfield imaging of the same field. For each treatment group, 5–10 mice were used; for each sample, five different fields in both primary spongiosa and secondary spongiosa were calculated.

**Fluorescence in situ hybridization**. The probe used for detection of 47S rRNA was prepared as previously described[104]. Briefly, the templates for the sense riboprobes were prepared by PCR from mouse genomic DNA with sense primer containing a T7 promoter (5′-GGGTAATAGGACTCACTATAGGGCGA). The primers for the initiation site of the 47S rRNA precursor were forward 5′-GCCTGTCACTTTCCTCCCTG and reverse 5′-GCCGAAATAAGGTGGCCCTC. The probe was labeled by Alexa Fluor 488 using a FISH Tag™ RNA Kit (Invitrogen). FISH was conducted according to the protocol described by Thomas Liehr[109] with modification. Paraffin-embedded bone tissue sections were deparaffinized and

subjected to a series of pre-hybridization treatments. Hybridization was performed using 1 μg/mL Alexa Fluor 488-labeled probe at 55 °C for 16 h. After successive post-hybridization washing, nuclei were counterstained with DAPI (Sigma). The sections were mounted with the ProLong Antifade Kit (Molecular Probes) and observed under a Zeiss LSM780 confocal microscope. FISH combined with immunofluorescence staining was performed as previously described[109]. Immunofluorescence staining of paraffin-embedded bone tissue sections was carried out before FISH. Both immunofluorescence staining and FISH were performed according to the same procedure described above. Images were acquired with a Zeiss LSM780 confocal microscope, and the intensity profiles were acquired with Image J software.

**Micro-CT analysis**. Mice were anesthetized by inhalation of 2.5% isoflurane (Abbott Laboratories, Abbott Park, IL) mixed with $O_2$ (1.5 L/min). For μCT analysis, mice femora were dissected free of soft tissue, fixed overnight in 10% formalin at 4 °C, and analyzed by high-resolution μCT (Skyscan 1172, Bruker MicroCT, Kontich, Belgium). The scanner was set at 65 kV, 153 μA, and a resolution of 9.0 μm/pixel. We used NRecon image reconstruction software, version 1.6 (Bruker MicroCT), CTAn data-analysis software, version 1.9 (Bruker MicroCT), and CTVol 3-dimensional model visualization software, version 2.0 (Bruker MicroCT) to analyze parameters of the trabecular bone in the metaphysis. To perform 3-dimensional histomorphometry analysis of trabecular bone, we selected the regions of interest from 1 mm below the distal epiphyseal growth plate and extended toward the distal direction for proximally 2 mm length. Trabecular bone was analyzed to determine the trabecular BV/TV, Tb. Th, Tb. N, and Tb. Sp. Femur length was analyzed by using CTAn data-analysis software, version 1.9 (Bruker MicroCT).

**Histochemistry, immunohistochemistry, and histomorphometric analysis**. The femora were resected and fixed in PBS (pH 7.4) containing 4% paraformaldehyde for 48 h, decalcified in 0.5 M EDTA (pH 7.4) at 4 °C, and embedded in paraffin. Four μm-thick longitudinally oriented sections of bone, including the metaphysis and diaphysis, were processed for Safarin O-fast green (SOFG) staining, hematoxylin–eosin (H&E), and immunohistochemical staining. Osteoclasts were stained for tartrate-resistant acid phosphatase (TRAP) and counterstained with methyl green. Osteoblasts were stained by OCN immunohistochemistry. All sections were observed using an Olympus BX51 microscope. Quantitative histomorphometry analyses were performed as described previously[95,110,111] in a blinded fashion using OsteoMeasure Software (OsteoMetrics, Inc., Decatur, GA, USA). The sample area selected for calculation was a 1 mm$^2$ area within the metaphyseal trabecular bone. Number of osteoblasts per bone perimeter (Ob.N/B. Pm), osteoblast surface per bone surface (Ob.S/BS), number of osteoclasts per bone perimeter (Oc.N/B. Pm), and osteoclast surface per bone surface (Oc.S/BS) in five randomly selected visual fields per specimen, in five specimens per mouse in each group were measured. Goldner's trichrome stain was performed using undecalcified bone tissue sections to show osteoid (red) and mineralized bone matrix (green). For dynamic histomorphometry, two sequential doses of calcein (8 mg/10 mL in sterile saline) were injected intraperitoneally at 2 and 8 days before euthanization.

For serum P1NP measurement, blood serum was collected by cardiac puncture immediately before sacrifice and stored at −80 °C until use. N-terminal propeptide of type I procollagen (P1NP) was measured using EIA kit (Immunodiagnostic Systems, AC-33F1) as described previously[95].

**Preparation of bone cell suspension from femoral metaphysis**. Cell suspension from femoral metaphysis was prepared as described previously[71]. Briefly, femoral bones were isolated from mice and dissected free of soft tissues. The epiphysis was gently removed using forceps and 4 mm-long metaphysis region was dissected and further processed. The isolated metaphysis was cut into small pieces with scissors and was digested with digestion buffer (2 mg/mL collagenase I and 2 mg/mL collagenase IV in phosphate-buffered saline [PBS]) for 30 min. Cells were then filtered

through 70 μm cell strainer and the supernatant were collected for further experiments.

**Cell sorting and flow cytometry analysis of bone vascular cells.** Flow cytometric analysis and FACS sorting of type-H vessels or CD144+ endothelial cells from femoral metaphysis were performed as previously described[82,112] with modifications. Briefly, the epiphysis was removed from the distal femora and proximal tibia, and only the metaphyseal region was processed. The bones were then crushed in ice-cold PBS with a mortar and pestle. Whole bone marrow was digested with collagenase A (Sigma, 11088793001, 2 mg/mL) and trypsin (2.5 mg/mL) in PBS at 37 °C for 20 min to obtain single-cell suspensions. ACK lysis buffer is used to remove red blood cells. For analysis and sorting of type-H vessels, after washing and filtration, equal numbers of cells were incubated for 45 min at 4 °C with Emcn antibody (Santa Cruz, sc-65495, 1:50) and APC-conjugated CD31 antibody (R&D SystemsEmcn, FAB3628A, 1:100). After washing, were stained with BV421-conjugated second antibody against Emcn (BioLegend, 405414, 1:50) for 45 min. For analysis of the senescent vascular endothelial cells, after washing and filtration, equal numbers of cells were incubated with SPiDER-βGal working solution (Cellular Senescence Detection Kit-SPiDER-βGal, G04, Dojindo Molecular Technologies, Inc.) for 30 min. After washing, cells were stained with CD144-BV421 (BioLegend, 138013, 1:100) for 45 min. For analysis of senescent vascular endothelial cells in *p16tdTom* mice, cells were stained with CD144-BV421 (BioLegend, 138013, 1:100) for 45 min. After washing, cells were analyzed using a BD LSRII flow cytometer.

**Single-cell ImageStream analysis.** In an ImageStreamX system, which combines the features of fluorescent microscopy and flow cytometry, cells in suspension pass through the instrument in single file, where transmitted light, scattered light, and emitted fluorescence are collected. This is accompanied by image analysis, which allows quantification of intensity, location, morphology, and population statistics within tens of thousands of cells per sample. Single-cell suspensions were prepared from femoral metaphysis using the approach described above. Cells were stained with BV421-labeled CD144 antibody (BioLegend, 138013, 1:100), PE-labeled HMGB1 antibody (BioLegend, 651404, 1:100), and Alex594-labeled Ki67 antibody (Abcam, ab216709) and subjected to ImageStreamX mark II (Amnis, Part of EMD milipore—Merck, Seattle, WA) according to the strategy previously described[81]. At least $1 \times 10^5$ cells were collected from each sample. Images were analyzed using IDEAS v6.1 software (Amnis, Part of EMD Millipore-Merck, Seattle, WA). Cells were gated for single cells using the area and aspect ratio features on the brightfield image.

**Quantitative real-time PCR.** Total RNA for qRT-PCR was extracted from the cultured or sorting cells using RNeasy Mini Kit (QIAGEN, 74014) according to the manufacturer's protocol. Complementary DNA (cDNA) was prepared with random primers using the SuperScript First-Strand Synthesis System (Invitrogen) and analyzed with SYBR GreenMaster Mix (QIAGEN) in the thermal cycler with two sets of primers specific for each targeted gene. Target-gene expression was normalized to glyceraldehyde 3-phosphate dehydrogenase (GAPDH) messenger RNA, and relative gene expression was assessed using the $2^{-\Delta\Delta CT}$ method. Primer sequences are provided in Supplementary Table 1.

**Expression profiling of angiogenesis factors in metaphysis bone tissue.** The bone extracts collected from femoral metaphysis were subjected to a proteome profiler array using a mouse angiogenesis array kit (R&D, ARY015), and the relative levels of 53 angiogenesis-related proteins were assessed simultaneously. The array procedure and data analysis were performed according to the manufacturer's instructions.

**Ex vivo metaphysis explant culture.** Immediately after sacrificing the mice, the femurs were dissected from 3-week-old *C57BL/6J* mice and dissected free of soft tissue. The epiphysis was gently removed using forceps and 4 mm-long metaphysis region was dissected and cut in half with scissor longitudinally. The metaphysis explants were washed gently with ice-cold PBS three times and cultured in serum-free DMEM supplemented with 2 mM L-glutamate at 37 °C for 24 h to adjust the explant to the ex vivo environment. Then the metaphysis explant was cultured in serum-free DMEM with MPS ($10^{-6}$ M), PTH ($5 \times 10^{-7}$ M), Alendronate ($10^{-5}$ M), MPS + RU486 ($10^{-6}$ M), or vehicle for 3 days. After treatment, the explants were washed with PBS and cultured for another 3 days to collect conditioned media (CM). The CM was subjected to centrifugation (2500 r.p.m. for 10 min at 4 °C), and the supernatant was aliquoted and stored at −80 °C until use. ANG concentration in the CM was measured by mouse ANG ELISA kit (Abcam, ab208349) according to the manufacturers' instruction.

**Osteoclast culture.** In vitro osteoclastogenesis assays were performed as described previously[18,112] with modifications. Briefly, mouse bone marrow cells were harvested from 4-week-old male *C57BL/6J* mice by flushing marrow cavity from both tibia and femur, filtered through 40 μm cell strainer, and cultured with alpha minimum essential medium (α-MEM) containing 15% fetal bovine serum (FBS), 100 U/mL streptomycin sulfate (Sigma-Aldrich) and 100 U/mL penicillin

(Sigma-Aldrich) in 10-cm culture dishes at 37 °C in 5% CO₂ humidified incubator for at least 24 h following established protocols. The adherent cells were discarded while the floating cells were cultured with macrophage colony-stimulating factor (M-CSF; R&D Systems) at 30 ng/mL for 48 h. To collect pure monocytes/macrophages, the cells are digested with Versene for ~4 min in 37 °C until most macrophages are de-associated. Then the cells are seeded in 24-well plate and cultured with 30 ng/mL M-CSF and 100 ng/mL RANKL (Abcam, ab129136) for 5 days to induce the formation of mature osteoclasts. For treatments, $10^{-6}$ M MPS with/without $10^{-6}$ M RU486 is added to culture medium at designated timepoints and cultured for 24 h. TRAP activity is measured using a commercial kit (Sigma-Aldrich). Total RNA was extracted, and the mRNA level of ANG was measured by qRT-PCR.

**Cell culture and siRNA treatment.** Cryopreserved human umbilical vein endothelial cells (HUVECs) were purchased from ATCC (PCS-100-013, Gaithersburg, MD). Cells were cultured in vascular cell basal medium (PCS-100-030, ATCC) supplemented with Endothelial Cell Growth Kit-BBE (PCS-100-040, ATCC), penicillin (50 U/mL), and streptomycin (50 μg/mL). Second to fourth passages of endothelial cells were used throughout. For siRNA treatment, endothelial cells (20–30% confluent) were transfected with siRNA using Lipofectamine RNAiMAX (Thermo Fisher, 13778-150) according to manufacturer's instructions. Scrambled (control) and PLXNB2 siRNAs (Thermo Fisher, Stealth RNAi siRNA) were used for experiments. Impedance measurements were performed 48 h after transfection.

**Cell proliferation assay.** HUVECs were seeded in six-well plate at 20% confluency in culture medium and incubated overnight at 37 °C. Upon treatment, cell culture medium was changed into vascular cell basal medium containing 1% FBS, and rhANG (1 mg/mL, kindly provided by Dr. Guo-Fu Hu) or individual CM media prepared as described above (1:1 with vascular cell basal medium containing 1% FBS) was added and cultured for 24 h. Cell proliferation was assayed using MTT assay kit (Roche, 11465007001) according to the manufacturer's instruction. MTT assays were performed in quadruplicates each time and repeated three times.

**Tube formation assay.** Growth factor reduced Matrigel (Corning, 356230) was added into 24-well culture plates and incubated at 37 °C for 45 min to solidify the gel. HUVECs were seeded in 20% confluency on polymerized Matrigel in plates. rhANG (1 μg/mL, kindly provided by Dr. Guo-Fu Hu) or individual CM media prepared as described above (1:1 with vascular cell basal medium containing 1% FBS) was added and cultured for 4 h. In the experiment of using neutralizing antibodies, 30 μg/mL of non-immune isotype IgG control, 26-2F (eBioscience, 14-9762-82), or mAb17 (kindly provided by Dr. Guo-Fu Hu) was added in the culture medium with CM prepared from control metaphysis explant culture (1:1 with vascular cell basal medium containing 1% FBS). In some experiments, Tube formation was observed under the microscopy, and the cumulative tube lengths were measured.

**Simultaneous Click-iT RNA imaging assay.** Click-iT RNA Imaging Assay (Life Technologies, C10329) was performed as described previously[113,114]. EU used in this assay is an alkyne-modified nucleoside. When EU is fed to cells, it can be actively incorporated into the cells. Then the modified RNA nucleoside can be detected with a corresponding azide-containing dye and the fluorescence intensity of EU reflects the amount of newly synthesized RNA. Briefly, cells were seeded on coverslips, provided with 1 mM EU in complete DMEM and pulsed for 8 and 16 h. Cells were then washed twice with PBS and fixed with 3.7% formaldehyde in PBS. Coverslips were washed once with PBS and permeabilized with 0.5% Triton X-100 for 15 min. Permeabilization buffer was washed away, and staining was performed with Click-iT reaction mixture containing the azide-conjugated Alexa Fluor 488 dye for 30 min in a wet and dark chamber at room temperature. Nuclei were counterstained with DAPI (Sigma). The sections were mounted with the ProLong Antifade Kit (Molecular Probes, Eugene, OR) and observed under a Zeiss LSM780 confocal microscope (Olympus America, Inc, Center Valley, PA).

**TUNEL assays.** To detect apoptosis cells, terminal deoxynucleotidyl transferase dUTP nick end labeling (TUNEL) assays were conducted using the TACS® 2 TdT-Fluor In Situ Apoptosis Detection Kit in Situ Cell Death Kit (R&D Systems, 4812-30-K) according to the manufacturer's instruction. For in vivo assays, formalin-fixed, OCT-embedded fresh frozen sections were used. TUNEL reaction mixture (50 μL) was added to the slides and incubated for 1 h at 37 °C in a humidified chamber in dark. Negative control was set up and processed with the same procedures. To detect apoptosis rate in cells, cells were seeded on coverslips. After treatment, cells were fixed with formalin, washed once with PBS, resuspended in 50 μL of TUNEL reaction mixture and incubated for 60 min at 37 °C in a humidified chamber in the dark. In both procedures after labeling procedure, the slides or coverslips were rinsed three times with PBS and mounted with cover glasses using DAPI and analyzed under fluorescence microscope.

**Western blot analysis.** Western blot analysis was conducted as previously described[115,116]. Briefly, cells were lysed in RIPA lysis buffer and harvested with a rubber policeman. 20 μg protein was separated by SDS–PAGE and electro-transferred

onto a nitrocellulose membrane. After blocking, the membrane was incubated with primary antibody at 4 °C overnight and then with secondary antibodies at RT for 1 h. Proteins were visualized with chemiluminescence detection reagents followed by autoradiography. The following antibodies were used: PLXNB2 (Proteintech, 10602-1-AP, 1:800), AKT (Cell Signaling, 40D4, 1:2000), p-AKT (Cell Signaling, 193H12, 1:1000), ERK1/2 (Cell Signaling, 137F5, 1:1000), p-ERK (Santa Cruz, sc-7383, 1:500), Lamin B1 (Santa Cruz, sc-374015, 1:500), Ki67 (Abcam, ab92742, 1:2000), p21 (Abcam, ab109520, 1:2000), GAPDH (Cell Signaling, 14C10, 1:1000), and β-Tubulin (Cell Signaling, 9F3, 1:1000).

**Statistics**. Data are presented as means ± standard errors of the mean. Unpaired, 2-tailed Student $t$-tests were used for comparisons between two groups. For multiple comparisons, one-way analysis of variance (ANOVA) with post hoc Tukey test was used. All data were normally distributed and had similar variation between groups. Statistical analysis was performed using SAS, version 9.3, software (SAS Institute, Inc., Cary, NC). $p < 0.05$ was deemed significant. All representative images of bones or cells were selected from at least three independent experiments with similar results unless indicated differently in the figure legend.

**Reporting summary**. Further information on research design is available in the Nature Research Reporting Summary linked to this article.

## Data availability

The data that support the findings of this study are available within the article and Supplementary Files or from the corresponding author upon reasonable request. Source data are provided with this paper.

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

## Acknowledgements

The authors sincerely thank Dr. Guo-Fu Hu (Tufts Medical Center) for kindly providing the ANG polyclonal antibody for immunofluorescence staining, recombinant human ANG, and mAb17 antibody for in vitro experiments. We also thank Gloria H. Su (Columbia University Medical Center) for kindly providing the *p16^f/f* mice. We acknowledge the assistance of Johns Hopkins Ross Flow Cytometry Core Facility (supported by NIH shared-instrument grant) and School of Medicine Microscope Facility (supported by NIH, S10OD016374). The authors also acknowledge the assistance of Rachel Box and Jenni Weems at The Johns Hopkins Department of Orthopedic Surgery Editorial Services for editing the manuscript. This work was supported by the National Institutes of Health grant R56 AG059578 to M.W.

## Author contributions

X. Liu. and M.W. designed the experiments. X. Liu. and Y.C. carried out most of the experiments; B.Y., G.L., Q.G., X. Lv. and W.S. helped with some experiments; X.L. and M.W. designed and drew the illustrations in Figs. 1a; 2g; 4a and 6a; P.G., G.F., and X.C. proofread the manuscript; M.W. supervised the experiments, analyzed results, and wrote the manuscript.

## Competing interests

The authors declare no competing interests.
