## [Peer Review File · Nature Communications]

Reviewers' comments:

Reviewer #1 (Remarks to the Author):

Review Draft 080919

This study aims to elucidate the effects of GCs on children implicating senescence in endothelial cells as a contributor of GC effects on bone. In a previous study the authors recognized senescence as a feature of puberty in long bone growth. In this previous study the authors described a reduced number of Nes-GFP mesenchymal cells and concluded enhanced senescence. In a slightly different Pred treatment regimen the authors show again enhanced senescence based on b-gal, HMGB1 and p16ink4A overall cell types.

While this is known, the authors show novel findings, that H-Type vessels are subject to senescence upon MPS treatment and this is accompanied by changed ANG expression and reduced 47S rRNA synthesis. The authors perform functional experiments to show that reduction of ANG affects bone integrity and that recombinant ANG rescues MPS effects on bone.

This is a very interesting study and sheds new light into the mechanisms of GC induced bone effects.

One major aspect is missing, that is the potential mechanism how GR regulates ANG expression or whether it rather indirectly regulates the number of cells with high and low ANG expression. The molecular mechanism of ANG activity could be further elucidated, e.g. whether it acts via PLXNB2. Further a more detailed analysis of cell numbers versus senescence and apoptosis and autophagy should be performed at least at some example. The effects of Cdh5Cre Ang KO mice should be also evaluated on the basal level, i.e. in vehicle treated mice.

Major points:

1. The authors should show, how ANG is regulated by Dex. They should show in the HUVECs, how mRNA, protein levels are reduced or not. Maybe Dex rather affects nuclear localization only? Furthermore, RU486 treatment should demonstrate specificity of the GR.
2. To authors should show whether Plexin-B2 that was recently suggested as a mediators of action is expressed in endothelial cells and bone cells and whether silencing e.g. in HUVECs abrogates the actions of recombinant ANG.
3. The authors should show the analysis of the senescence markers also after after one week of treatment.
4. The authors should show directly the senescence marker expression of NesGFP+ cells.
5. The authors should show whether TRAP+ and F4/80+ cells change in absolute numbers.
6. Similarly absolute numbers of Emcn positive cells and CD144 positive cells should be shown.
7. It would be very interesting if the authors would demonstrate the rate of apoptosis, autophagy for the different cell types in comparison to the senescence.
8. The authors should comment on the differences of vehicle treated animals at 1 and 3 weeks in absolute numbers of Osx+, OCN+ and Emcn/CD31 cells per area.
9. It would strengthen the findings, if the authors could comment, whether 47S rRNA signal is also reduced in Osx+, Ocn+ and other cells in the bone.
10. The authors should show, whether in after siRNA treatment against ANG, Dex can still enhance b-Gal activity.
11. In Figure 7, the vehicle treated mice have to be shown and all parameters reflected (including

microCT analysis). Here ANG dependent and ANG independent effects of MPS have to be elaborated.

12. The effect of rhANG treatment should be also shown on osteoblasts, osteoclasts and osteocytes.

13. The tail length data are interesting. The authors should comment also whether this has effects on the growth plate.

Minor points

1. The studies of Rauch et al. Cell Metabolism 2010 and Weinstein et al. 2017 should be mentioned in the introduction.

2. The difference of ANG immunostaining in the secondary spongiosa is hardly visible. Also the statement that Ang is represented in "pink color" in Fig. 4C, D is not appropriate.

3. The description in the text for Fig. 6A (3rd and 4th row) is misleading. There are only two rows in A. The other rows are B and C.

Reviewer #2 (Remarks to the Author):

In this paper, Chai et al. link glucocorticoid (GC) treatment in mice to cellular senescence of vascular endothelial cells. At a mechanistic level, they show that the effects of GCs are mediated, at least in part, through reduced angiogenin (ANG) expression. ANG treatment blocks at least part of the effects of GC treatment on bone in growing mice.

Overall, the studies are done well and the results support the conclusions. I do have some issues for the authors to address:

1. The use of antibodies to mouse p16INK4a is problematic. For example, the Santa Cruz antibody used in this paper (sc-1661) has been shown to detect a persistent signal by IHC even after p16INK4a knockdown (Sawicka et al. PLOS One 8:e53313, 2013). The Abcam antibody used has also not been well validated, as the company does not provide negative controls in their validation. Because so much of the paper uses antibodies to mouse p16INK4a, the authors need to provide convincing data regarding the specificity of the p16INK4a antibodies they used, and also indicate in each experiment exactly which antibody (Santa Cruz or Abcam) is being used. It should be noted that it is in part due to this problem with mouse p16INK4a antibodies that the Sharpless group developed reporter mice for p16INK4a expression (doi: 10.1073/pnas.1818313116).
2. In the experiments using Tam, appropriate controls should be provided demonstrating no independent effects of Tam on the outcome parameters being assessed.
3. The strains of the mice used should be specified. Were all mice in the same background?

Reviewer #3 (Remarks to the Author):

In this study, Chai et al., demonstrate that the anti-angiogenic effect of glucocorticoids (GC) treatment in bone is mediated by angiogenin (ANG) loss. They suggest ANG expressed by type-H vessels promotes rRNA synthesis and proliferation of endothelial cells. GC treatment led to ANG loss and promoted endothelial cell senescence.

Numbers of evidence, in particular, Weinstein et al., Aging cell, 2010 showed that GCs inhibit angiogenesis of endothelial cells in bone. Studies by Hu lab established ANG function in endothelial cells through rRNA synthesis. The main finding of the current study, GCs regulation of ANG in bone endothelial cells needs detailed investigation. The molecular mechanism behind GCs control of ANG needs to be explained.

Besides, the following points in the data need to be addressed to support the findings.

1. ANG staining: Nuclear and cytoplasmic localisations are not visible in images. It would also be more comprehensible for the readers to understand the status if ANG stainings are always co-

stained with blood vessel markers.

2. Transgenic reporter mouse line, Nestin-GFP was used to label blood vessels in the metaphysis. It marks mesenchymal cells as well. The endothelial-specific expression should be illustrated using Cdh5-iCre driven reporter or any other reporter mice to label bone blood vessels and ANG localisation in type-H cells and not in other mesenchymal population.
3. Authors suggest EC specific expression of ANG. However, it is also essential to check ANG is not expressed on other cell types such as mesenchymal and haematopoietic lineage.
4. In Fig 2a, is beta-gal localisation nuclear?
5. In Fig 3D, both type-H & L cells seem to be reduced. It would be useful to provide quantifications for type-L cells in this experiment and also in transgenic animals (both in the presence and absence of GCs).
6. Because ANG loss is downstream of GC, whether authors have tested Ang loss of function mouse models to understand they display a bone phenotype similar to GC treated animals.
7. Why some experiments were done with DEX, use prednisolone in all. If there is a reason, explain.
8. In fig 4, what is 47S rRNA? The pre-ribosomal RNA is 45S
9. No cytoplasmic ANG in figure 5
10. What is EU labelling?
11. ANG role in myeloid progenitors has been reported (Goncalves et al., 2016 Cell). The authors demonstrate that osteoclast are not affected. The reason could be explained.
12. To understand the endothelial-specific expression pattern of P16 ink4a and HMGB1 in many images, they need to be co-stained with endothelial marker/reporter in all the images.
13. Fig. 7 needs appropriate controls. The phenotype of blood vessels in transgenic mice in the presence and absence of GCs should be shown.
14. The vascular phenotype of p16 iKO should be defined
15. Fig 2h, what is cd14?

Reviewer #4 (Remarks to the Author):

This manuscript reports that angiogenin insufficiency in bone vascular cells is a reason for glucocorticoids (GC)-induced osteoporosis. Convincing data was presented to show that GC treatment in young mice suppresses nucleolar expression of ANG selectively in endothelial cells comprising type-H vessels in the metaphysis and that ANG-mediated rRNA transcription is an underlying mechanism for endothelial cell senescence. The authors also demonstrated that systemic treatment with ANG rescues GC-induced growth retardation accompanied with normalization of rRNA transcription and endothelial cell proliferation. These data identified a novel GC target with significant therapeutic implications to lessen the adverse effects associated with long term use of GC. My major comment, which is not a concern, is that cell autonomous and non-cell autonomous function of ANG in bone endothelial cells of the spongiosa regions are not clear. The authors presented some evidence that endogenous ANG in endothelial cells played a role but ANG is also secreted by other types of cells such as osteoblasts. Are those ANG, released into the micro environment, not taken up by endothelial cells to compensate the loss? Are other cells affected by GC in term of ANG secretion? This needs to be addressed, at least discussed, in light of the concern that the conclusion of endothelial cells being the primary GC target cells was not sufficiently supported (see below).

Other comments

1. Figure 1J, Nestin positive cells were decreased after MPS treatment. Are those endothelial cells or osteoprogenitor cells? Are endothelial cells in the primary and secondary spongiosa regions of metaphysis bone nestin positive?
2. Figure 1G shows that there is an apparently lower frequency of HMGB1 positive cells upon MPS treatment but the overall intensity of HMGB1 staining is more or less the same in positive cells even though nuclear localization of HMGB1 is lost. However, Figure 2O shows a decrease in HMGB1 expression in CD144 positive cells but no change in the frequency of positive cells. What is the reason for this discrepancy? Are all HMGB1 positive cells in Figure 1G CD144 positive? Double IF should be done to ensure if only endothelial cells are senescent after MPS treatment. The conclusion that GC treatment primarily causes cellular senescence of vascular endothelial cells in the metaphysis of glowing bone is not sufficiently supported.
3. Figure 3. Do you see any difference in L-type blood vessels?

4. Fig. 5G. How many ANG-specific siRNA were tested? Do you have mRNA and protein data on the knockdown efficiency?
5. Figure 8H and 8I. In the text and figure legend, Figure 8H and 8I are supposed to be *Osx* positive cells, however, the figures actually show beta-Gal positive vessels.
6. Page 18, Gloria Su appeared to be a coauthor and should not be acknowledged for providing the mice, it will be more appropriate to give a citation of these mice.
7. The ANG product used in systemic administration was a 15 amino acid synthetic peptide near the center of human ANG (Thermo Fisher, PEP-1464, page 19, line 5 from bottom) rather than the entire ANG protein. If the authors have not examined the activity of recombinant ANG protein, it should be done before the conclusion that it is the ANG-ribosome biogenesis pathway that is responsible for the therapeutic activity of ANG against GC-induced osteoporosis. It is hard to believe that a 15 amino acid peptide of ANG from the center part of the protein will restore ribosome biogenesis. I hope that it is only a typo when the corresponding author wrote the manuscript.
8. Page 20, line 8. Thermo Fisher 4390843 is a control siRNA not ANG-specific siRNA, please correct.

Point-by-point response to reviewers' comments

We would like to thank the reviewers for their thoughtful and constructive comments regarding our manuscript. We have addressed all of the questions and concerns brought forth through additional experimentation and clarification. To aid in readability, we have made the modified text in blue font to distinguish from the black original text. The following responses have been prepared to address all of the reviewers' comments in a point-by-point fashion.

Response to comments from Reviewer #1:

General Comments:

This study aims to elucidate the effects of GCs on children implicating senescence in endothelial cells as a contributor of GC effects on bone. In a previous study the authors recognized senescence as a feature of puberty in long bone growth. In this previous study the authors described a reduced number of NesGFP mesenchymal cells and concluded enhanced senescence. In a slightly different Pred treatment regimen the authors show again enhanced senescence based on bGal, HMGB1 and p16ink4A overall cell types. While this is known, the authors show novel findings, that H-Type vessels are subject to senescence upon MPS treatment and this is accompanied by changed ANG expression and reduced 47S rRNA synthesis. The authors perform functional experiments to show that reduction of ANG affects bone integrity and that recombinant ANG rescues MPS effects on bone. This is a very interesting study and sheds new light into the mechanisms of GC induced bone effects.

One major aspect is missing, that is the potential mechanism how GR regulates ANG expression or whether it rather indirectly regulates the number of cells with high and low ANG expression. The molecular mechanism of ANG activity could be further elucidated, e.g. whether it acts via PLXNB2. Further a more detailed analysis of cell numbers versus senescence and apoptosis and autophagy should be performed at least at some example. The effects of Cdh5Cre Ang KO mice should be also evaluated on the basal level, i.e. in vehicle treated mice.

Response: We thank Reviewer #1's overall positive comments and constructive suggestions. In the revised manuscript, we have conducted detailed characterization on how glucocorticoids (GC)/glucocorticoid receptor (GR) regulates ANG expression in metaphysis of growing bone. Our new result from immunofluorescence analysis using a specific ANG antibody (kindly provided by Dr. Guo-fu Hu's laboratory in Tufts Medical Center) clearly demonstrate that ANG is almost exclusively expressed in RANK⁺TRAP⁺ osteoclasts in metaphysis, particularly in the primary spongiosa adjacent to growth plate. Importantly, GC treatment led to dramatic reductions in both the number of the osteoclasts in this region and the expression of ANG in osteoclasts (**Figure 4B-4J** and **Figure S6**). Moreover, we established an *ex vivo* model system, in which endothelial cells were cultured with conditioned medium (CM) prepared from femoral metaphysis explant culture. Consistent with the *in vivo* finding, our *ex vivo* results show that there was a dramatic reduction in ANG secretion in response to GC treatment. As a result, there were declines in the proliferation and angiogenesis of endothelial cells that were cultured with the CM from MPS-treated explants relative to the CM from control explants (**Figure 5**).

Osteoclasts at the bone/cartilage interface (primary spongiosa) were recently termed as vessel-associated osteoclast subtype with no bone-resorbing activity. This distinct subset of osteoclasts is closely associated with type H vessels in metaphysis and is essential for maintaining angiogenesis and osteogenesis in this region (*Nat Cell Biol* 2019; 21:430-441). Our result that the ANG-expressing osteoclasts are closely associated with Emcn⁺ vascular endothelial cells (**Figure 4K**) agrees with this finding. More importantly, our work provides a molecular mechanism for the osteoclast-vascular crosstalk in this specific skeletal region. Our new *in vivo* and *in vitro* results show that ANG-Plexin B2 (PLXNB2) axis mediates the osteoclast-vascular interplay, which is essential to maintain vascular endothelial cells from senescence in metaphysis of growing bone. Specifically, we found that PLXNB2, a well-recognized ANG receptor, is strongly expressed in vascular endothelial cells in metaphysis of growing bone (**Figure 6A**). Moreover, knockdown of PLXNB2 in endothelial cells was sufficient to inhibit ANG nuclear translocation and rRNA transcription, resulting in cellular senescence and impaired angiogenic activity (**Figure 6E-6K**). In contrast, administering a recombinant ANG (rh-ANG) significantly attenuated endothelial cell senescence, improved angiogenesis-osteogenesis, and successfully protects growing bone from the negative effects of GC treatment on bone growth and bone formation (**Figure 7**). Collectively, our new results suggest that osteoclast-derived ANG maintains the osteogenesis-coupled type H vessels in metaphysis of growing skeleton via an ANG/PLXNB2-rRNA transcription signaling pathway; and inhibition of this pathway by GCs leads to vascular cell senescence and resultant impaired angiogenesis with osteogenesis (illustrated in **Figure 8**). Because of the critical new findings described above, we have made substantial changes to the manuscript and changed the title of the manuscript into “Osteoclasts Protect Bone Blood Vessels Against Senescence through the Angiogenin/Plexin-B2 Axis”.

Specific Comments:

1. The authors should show, how ANG is regulated by Dex. They should show in the HUVECs, how mRNA, protein levels are reduced or not. Maybe Dex rather affects nuclear localization only? Furthermore, RU486 treatment should demonstrate specificity of the GR.

Response: In the past year, we have conducted a panel of new experiments to determine the main source of ANG in growing bone, and found that vascular endothelial cells in metaphysis is a main target of ANG rather than serving as a source of it. Our new results demonstrate that ANG from osteoclasts acts on the closely associated vascular endothelial cells in metaphysis via a non-cell autonomous mechanism. Please also see our response to the General Comments of this reviewer.

Our new *in vivo* and *ex vivo* data show that ANG is primarily secreted by RANK⁺TRAP⁺ osteoclasts in metaphysis, and GC treatment significantly inhibited the production of ANG from osteoclasts (**Figure 4** and **5**). As the reviewer suggested, we have added RU486 in the MPS-treated femoral metaphysis explant culture in our *ex vivo* experiment. Our result showed that MPS treatment significantly inhibited the secretion of ANG from osteoclasts, and addition of a GC receptor (GR) antagonist RU486 restored the production of ANG that was inhibited by MPS (**Figure 5B**). Therefore, GC directly inhibited ANG expression in osteoclasts through activating GR. Whether GR interacts with other transcription factors or coregulators to repress ANG gene transcription remains unknown and is an interesting topic for future study. This point has been added in Discussion section (Line 436-442).

2. The authors should show whether Plexin-B2 that was recently suggested as a mediator of action is expressed in endothelial cells and bone cells and whether silencing e.g. in HUVECs abrogates the actions of recombinant ANG.

Response: As the reviewer suggested, we examined whether PLXNB2 is a key mediator for the function of ANG on endothelial cells and osteoprogenitor cells. We found that most of the Emcn+ vascular endothelial cells (approximately 83.43±4.57%) express PLXNB2, whereas only 5.76±3.89% of Osx+ osteoprogenitors have PLXNB2 expression in metaphysis (**Figure 6A**), indicating that vascular endothelial cells are a major ANG targeting cell type. Our *in vitro* experiments consistently show that knockdown of PLXNB2 from endothelial cells markedly inhibited ANG nuclear translocation, rRNA transcription, and cell proliferation, resulting in endothelial cell senescence and impaired angiogenesis (**Figure 6E-6K**). Therefore, ANG/PLXNB2 with its downstream rRNA synthesis is a key signaling pathway to maintain endothelial cell proliferation and protect cells from cellular senescence. Please also see our response to the General Comments raised by this reviewer. This point has been added in the Discussion section (Line 444-448,; Line 465-474).

3. The authors should show the analysis of the senescence markers also after one week of treatment.

Response: As the reviewer suggested, we have examined cellular senescence at metaphysis using a senescence reporter mouse strain *p16^{tdTom}* (*Proc Natl Acad Sci U S A.* 2019; 116: 2603-2611) at different time points after MPS treatment. tdTom⁺ cells started to increase in metaphysis of long bone as early as one week, peaked at two weeks, and started to decline at three weeks after MPS treatment (**Figure 2A-2F**). This point has been added in the Discussion section (Line 389-392).

4. The authors should show directly the senescence marker expression of NesGFP+ cells.

Response: In the initial submission of this manuscript, we used nestin deficiency as a marker of senescent cells because we previously found that loss of nestin expression represents a feature of cellular senescence in metaphysis of long bone (*Nat Commun* 2017; 8:1312). In the revised manuscript, we have included a newly established senescence reporter mouse strain *p16^{tdTom}*, in which tandem-dimer Tomato (tdTom) was knocked into exon 1α of the *p16^{INK4a}* locus to enable the identification of *p16^{INK4a}*-activated cells (tdTom⁺) at the single-cell level (*Proc Natl Acad Sci U S A.* 2019; 116: 2603-2611). We have therefore included a new set of data demonstrating the dynamics of cellular senescence at metaphysis in response to GC treatment (**Figure 2A-2F**). The old data using Nestin-GFP as a maker has therefore been removed from the revised manuscript.

5. The authors should show whether TRAP+ and F4/80+ cells change in absolute numbers.

Response: As the reviewer suggested, we have performed TRAP, VPP3, and F4/80 staining. The result shows that the number of F4/80+ cells was unchanged, but the numbers of TRAP⁺ and VPP3⁺ osteoclasts were reduced after GC treatment. The representative images and the statistics have been included in **Figure S6**.

6. Similarly absolute numbers of Emcn positive cells and CD144 positive cells should be shown.

Response: Percentage of Emcn positive cells and CD144 positive cells in response to MPS treatment have been calculated from flow cytometry data, and the results have been included in **Figure S2**.

7. It would be very interesting if the authors would demonstrate the rate of apoptosis, autophagy for the different cell types in comparison to the senescence.

Response: As suggested, we have performed TUNEL assay using bone tissue sections. Our data show that MPS treatment indeed induces increased cell apoptosis in metaphysis of long bone. We have also assessed the apoptotic cells in different cell populations. We detected increased number of TUNEL⁺ cells in the Osx⁺ osteoprogenitor cell population but unchanged TUNEL⁺ cells in Emcn⁺ vascular endothelial cells in metaphysis in MPS-treated vs. vehicle-treated mice (**Figure S5A-S5C**). As we found that inhibited ANG/PLXNB2 signaling leads to endothelial cell senescence, we have also examined whether inhibiting ANG/PLXNB2 signaling also causes endothelial cell apoptosis *in vitro*. Knockdown of PLXNB2 in endothelial cells did not induce cell apoptosis as detected by TUNEL assays (**Figure S7**) but significantly induced cell senescence (**Figure 6I**). These results suggest that, in response to GC treatment, vascular endothelial cells primarily undergo senescence, whereas osteoblast lineage may have multiple cell fate changes, such as cell apoptosis, cell senescence, and others (*Weinstein et al. J Clin Invest 1998; O'Brien et al. Endocrinology 2004; Li et al. PLoS One 2012; Li et al. Cell Death Dis 2013*). GC treatment induces autophagy in osteocytes, which is required for osteocyte death. It will be informative in the future to test whether GC treatment stimulates autophagy in vascular endothelial cells in metaphysis and how autophagy plays a role in the senescence of this cell lineage in response to GCs. This important point has been added in the Discussion section (Line455-463).

8. The authors should comment on the differences of vehicle treated animals at 1 and 3 weeks in absolute numbers of Osx+, OCN+ and Emcn/CD31 cells per area.

Response: This is an important point that we should have given an explanation in our initial submission. In this figure (new **Figure 3A-3C** and **Figure 3G-3I**), we used 3-week-old mice, and the mice were treated with MPS or vehicle for 1 and 3 weeks. Therefore, the ages of the mice at sacrifice are different in these 2 treatment groups (4- and 6-week-old, respectively). Given that bone growth was fast during early puberty periods (2–5 weeks of age) and gradually decelerates during late puberty (5–8 weeks of age) (*Li et al. Nature Communications 2017*), it is reasonable that Emcn/CD31 vascular cells and osteoprogenitor cells are more abundant in the 1-week treatment group relative to 3-week treatment group. Our data is consistent with the recent several reports that angiogenesis and the coupled osteogenesis are abundant in growing young mice and gradually decline with age (*Kusumbe et al. Nature 2014; Romeo et al. Nature Cell Biology 2019*). Explanation on this point has been added in the revised manuscript (Line 195-200; 384-386).

9. It would strengthen the findings, if the authors could comment, whether 47S rRNA signal is also reduced in Osx+, Ocn+ and other cells in the bone.

Response: The FISH assay of bone tissue sections shows that vascular endothelial cells in metaphysis have strong 47S rRNA signal in vehicle-treated mice but significant reduced after MPS treatment (**Figure 6B** and **6C**). We note that 47S rRNA signal was also detected in cells other than endothelial cells in metaphysis. Given that osteoprogenitor cells are very abundant in metaphysis and highly proliferative in growing skeleton, we postulate that this type of cells may also have active rRNA transcription. However, the signal was not significantly reduced in the non-endothelial cells upon MPS treatment (**Figure 6D**). This result is consistent with our finding that only a small portion of *Osx*⁺ osteoprogenitor cells express ANG receptor PLXNB2, suggesting that osteoprogenitors in metaphysis may not be a main direct target of ANG. This important point has been discussed in the revised manuscript Line 444-448).

10. The authors should show, whether in after siRNA treatment against ANG, Dex can still enhance b-Gal activity.

Response: Our new data demonstrate that RANK⁺TRAP⁺ osteoclasts but not vascular endothelial cells in metaphysis are the primarily source of ANG. Because of this new finding, the studies using siRNA treatment against ANG in endothelial cells have been removed. Our new *in vitro* data show that knockdown of ANG receptor PLXNB2 in endothelial cells using siRNA is sufficient to induce cellular senescence and adding rhANG failed to inhibit endothelial cell senescence (**Figure 6I**). The results suggest that the ANG-PLXNB2 signaling is essential to protect vascular endothelial cells against cellular senescence.

11. In Figure 7, the vehicle treated mice have to be shown and all parameters reflected (including microCT analysis). Here ANG dependent and ANG independent effects of MPS have to be elaborated.

Response: We have added the group of vehicle-treated mice as suggested. We found that rhANG only moderately changed angiogenesis/osteogenesis and the bone parameters in vehicle-treated mice although it has significant effects on alleviating GC-induced bone loss and femoral length (**Figure 7**).

12. The effect of rhANG treatment should be also shown on osteoblasts, osteoclasts and osteocytes.

Response: As suggested, we have evaluated the changes of different cell types in metaphysis of rhANG-treated mice. The results show that the number of osterix⁺ osteoprogenitors was also increased in rhANG co-treated mice relative to MPS alone treated mice (**Figure 7E** and **7F**). rh-ANG partially restored the number of osteocytes (**Figure S8A** and **S8B**) but did not improve the number of Vpp3⁺ osteoclasts (**Figure S8C** and **S8D**) that were impaired by MPS treatment.

13. The tail length data are interesting. The authors should comment also whether this has effects on the growth plate.

Response: We have assessed the changes in growth plate in response to rhANG treatment. H & E staining of the metaphysis region shows that while MPS reduced the height of proliferation zone and hypotrophy zone, co-treatment of rhANG significantly alleviated the changes (**Figure S9C-**

S9F). The result suggests that diminished type H vessels in primary spongiosa induced by GC also leads to the growth plate impairment for growth retardation during childhood.

14. The studies of Rauch et al. Cell Metabolism 2010 and Weinstein et al. 2017 should be mentioned in the introduction.

Response: These two studies have been acknowledged and cited (Line 43, Ref 24; Line 45, Ref 34).

15. The difference of ANG immunostaining in the secondary spongiosa is hardly visible. Also the statement that ANG is represented in “pink color” in Fig. 4C, D is not appropriate.

Response: We are sorry for the low-resolution image provided regarding the ANG staining. We have repeated the immunofluorescence staining of ANG using a new specific antibody, and the data now clearly shows that ANG is strongly expressed in osteoclastic cells at primary spongiosa (**Figure 4B-4K**), and the ANG⁺ cells are almost undetectable in secondary spongiosa in vehicle-treated mice. A few scattered ANG⁺ cells were also found in diaphyseal bone marrow, but MPS treatment did not change the number of these cells (**Figure 4D** and **4F**), indicating a location-specific ANG expression patterns in bone.

16. The description in the text for Fig. 6A (3rd and 4th row) is misleading. There are only two rows in A. The other rows are B and C.

Response: The description in the text for this figure has been corrected (new **Figure 6**).

Response to comments from Reviewer #2:

General Comments:

In this paper, Chai et al. link glucocorticoid (GC) treatment in mice to cellular senescence of vascular endothelial cells. At a mechanistic level, they show that the effects of GCs are mediated, at least in part, through reduced angiogenin (ANG) expression. ANG treatment blocks at least part of the effects of GC treatment on bone in growing mice. Overall, the studies are done well and the results support the conclusions.

Specific Comments:

1. The use of antibodies to mouse p16INK4a is problematic. For example, the Santa Cruz antibody used in this paper (sc-1661) has been shown to detect a persistent signal by IHC even after p16INK4a knockdown (Sawicka et al. PLOS One 8: e53313, 2013). The Abcam antibody used has also not been well validated, as the company does not provide negative controls in their validation. Because so much of the paper uses antibodies to mouse p16INK4a, the authors need to provide convincing data regarding the specificity of the p16INK4a antibodies they used, and also indicate in each experiment exactly which antibody (Santa Cruz or Abcam) is being used. It should be noted that it is in part due to this problem with mouse p16INK4a antibodies that the Sharpless group developed reporter mice for p16INK4a expression (doi:10.1073/pnas.1818313116).

Response: We sincerely appreciate the valuable comment. To validate the senescence phenotype of bone vascular cells in response to GC treatment, we have acquired the murine senescence reporter strain $p16^{tdTom}$ (generated by Sharpless group) from Jackson Lab. $tdTom^+$ cells started to increase in metaphysis of long bone as early as one week, peaked at two weeks, and started to decline at three weeks after MPS treatment (**Figure 1D-1F**). Consistent results have been obtained from flow cytometry analysis of the $tdTom^+$ cells from metaphysis (**Figure 1A-1C**). The new set of data, consistent with the results from the SA- β Gal staining (**Figure 1G-1I**) and HMGB1 immunohistochemical analysis (**Figure 1J-1L**), suggest that GC induces cellular senescence in metaphysis of growing long bone. Using the $p16^{tdTom}$ mice, we have further identified that vascular endothelial cells and osteoprogenitor cells are main cell lineages undergoing senescence (**Figure 2A-2F**). As the reviewer pointed out, we recognize that the p16INK4a antibody used in our initial submission have specificity issue. In the revised manuscript, we have replaced all the p16INK4a immunostaining data with our new data from the $p16^{tdTom}$ mice.

In addition, in the past year, we have conducted a panel of new *in vivo*, *ex vivo*, and *in vitro* experiments and worked out the key mechanisms underlying glucocorticoid (GC)-induced vascular cell senescence in metaphysis of long bone. Specifically, our new data reveal that ANG produced from osteoclasts acts on neighboring vascular cells via a non-cell autonomous manner. ANG maintains the vascular endothelial cells from senescence through Plexin-B2-mediated rRNA transcription, whereas GC treatment inhibits ANG secretion from osteoclasts, resulting in impaired endothelial cell rRNA transcription and subsequent cellular senescence. Because of these critical new findings, we have made substantial changes to the manuscript and changed the title of the manuscript into “Osteoclasts Protect Bone Blood Vessels Against Senescence through the Angiogenin/Plexin-B2 Axis”.

2. In the experiments using Tam, appropriate controls should be provided demonstrating no independent effects of Tam on the outcome parameters being assessed.

Response: In the experiments using Tamoxifen (**Figure 3J-3O**), both $p16^{lox/lox}$ mice (WT littermates) and $Cdh5-Cre^{ERT2}; p16^{lox/lox}$ mice were actually treated with tamoxifen in all the treatment groups in order to avoid the effect of Tamoxifen on the outcome measures. The detailed treatment information has been included in the Methods section (Line504-508) and Figure Legend 3J-3O (Line 1120-1125).

3. The strains of the mice used should be specified. Were all mice in the same background?

Response: We are sorry for not clearly describing the mouse background in our initial submission. We used *BALB/c* mice in most of the experiments testing the phenotypic changes after MPS treatment. We also repeated some of the key experiments, for example **Figure 1G-1L, 2G-2I, 4B-4F**, using the *C57BL/6* mice. Since we obtained similar results for the key outcome measures as did in *BALB/c* mice, we did not present the data from the *C57BL/6* mice. Genetic-modified mice ($Cdh5-Cre^{ERT2}; p16^{lox/lox}$ (p16-iKO) mice and $p16^{lox/lox}$ (WT) mice) are in *C57BL/6* background. Detailed description on the background of the mice have been added in Methods section in the revised manuscript (Line 495, 499, 503).

Response to comments from Reviewer #3:

General Comments:

In this study, Chai et al., demonstrate that the anti-angiogenic effect of glucocorticoids (GC) treatment in bone is mediated by angiogenin (ANG) loss. They suggest ANG expressed by type-H vessels promotes rRNA synthesis and proliferation of endothelial cells. GC treatment led to ANG loss and promoted endothelial cell senescence. Numbers of evidence, in particular, Weinstein et al., Aging cell, 2010 showed that GCs inhibit angiogenesis of endothelial cells in bone. Studies by Hu lab established ANG function in endothelial cells through rRNA synthesis. The main finding of the current study, GCs regulation of ANG in bone endothelial cells needs detailed investigation. The molecular mechanism behind GCs control of ANG needs to be explained. Besides, the following points in the data need to be addressed to support the findings.

Response: We thank Reviewer #3 for the insightful comments. To address the concern on how glucocorticoids (GC) regulates ANG signaling in endothelial cells, we have conducted a panel of new *in vivo*, *ex vivo*, and *in vitro* experiments. Immunofluorescence analysis of femoral bone tissue sections using a new specific ANG antibody (kindly provided by Dr. Guo-fu Hu's laboratory in Tufts Medical Center) reveals that ANG is almost exclusively expressed in RANK⁺TRAP⁺ osteoclasts in metaphysis, particularly in the primary spongiosa adjacent to growth plate. Importantly, GC treatment led to dramatic reductions in both the number of the osteoclasts in this region and the expression of ANG in osteoclasts (**Figure 4B-4J** and **Figure S6**). Moreover, we established an *ex vivo* model system, in which endothelial cells were cultured with conditioned medium (CM) prepared from femoral metaphysis explant culture. Consistent with the *in vivo* finding, our *ex vivo* results show that there was a dramatic reduction in ANG secretion in response to GC treatment. As a result, there were declines in the proliferation and angiogenesis of endothelial cells that were cultured with the CM from MPS-treated explants relative to the CM from control explants (**Figure 5**).

Osteoclasts at the bone/cartilage interface (primary spongiosa) were recently termed as vessel-associated osteoclast subtype with no bone-resorbing activity. This distinct subset of osteoclasts is closely associated with type H vessels in metaphysis and is essential for maintaining angiogenesis and osteogenesis in this region (*Nat Cell Biol* 2019; 21:430-441). Our result that the ANG-expressing osteoclasts are closely associated with Emcn⁺ vascular endothelial cells (**Figure 4K**) agrees with this finding. More importantly, our work provides a molecular mechanism for the osteoclast-vascular crosstalk in this specific skeletal region. Our new *in vivo* and *in vitro* results show that ANG-Plexin B2 (PLXNB2) axis mediates the osteoclast-vascular interplay, which is essential to maintain vascular endothelial cells from senescence in metaphysis of growing bone. Specifically, we found that PLXNB2, a well-recognized ANG receptor, is strongly expressed in vascular endothelial cells in metaphysis of growing bone (**Figure 6A**). Moreover, knockdown of PLXNB2 in endothelial cells was sufficient to inhibit ANG nuclear translocation and rRNA transcription, resulting in cellular senescence and impaired angiogenic activity (**Figure 6E-6K**). In contrast, administering a recombinant ANG (rh-ANG) significantly attenuated endothelial cell senescence, improved angiogenesis-osteogenesis, and successfully protects growing bone from the negative effects of GC treatment on bone growth and bone formation (**Figure 7**). Collectively, our new results suggest that osteoclast-derived ANG maintains the osteogenesis-coupled type H

vessels in metaphysis of growing skeleton via an ANG/PLXNB2-rRNA transcription signaling pathway; and inhibition of this pathway by GCs leads to vascular cell senescence and resultant impaired angiogenesis with osteogenesis (illustrated in **Figure 8**). Because of the critical new findings described above, we have made substantial changes to the manuscript and changed the title of the manuscript into “Osteoclasts Protect Bone Blood Vessels Against Senescence through the Angiogenin/Plexin-B2 Axis”.

Specific Comments:

1. ANG staining: Nuclear and cytoplasmic localisations are not visible in images. It would also be more comprehensible for the readers to understand the status if ANG stainings are always co-stained with blood vessel markers.
&
3. Authors suggest EC specific expression of ANG. However, it is also essential to check ANG is not expressed on other cell types such as mesenchymal and haematopoietic lineage.

Response: We are sorry for not giving a clear image of ANG staining in the initial submission. In the past year, we have conducted a panel of new experiments to determine the main source of ANG in growing bone, and found that vascular endothelial cells in metaphysis is a main target of ANG rather than serving as a source of it. Specifically, our new data showed that ANG is almost exclusively expressed in RANK⁺TRAP⁺ osteoclasts in primary spongiosa adjacent to growth plate, and GC treatment reduced the expression of ANG in osteoclasts (**Figure 4B-4K**).

We further reveal that ANG from osteoclasts mainly acts on the nearby vascular endothelial cells in metaphysis via a non-cell autonomous mechanism (**Figure 4-6**). ANG maintains the vascular endothelial cells from senescence through Plexin-B2-mediated rRNA transcription, whereas GC treatment inhibits ANG secretion from osteoclasts, resulting in impaired endothelial cell rRNA transcription and subsequent cellular senescence. We have also explored other ANG-target cells in addition to endothelial cells. Our data show that most vascular endothelial cells have PLXNB2 expression, but only a small portion of Osx⁺ osteoprogenitor cells are PLXNB2-positive (**Figure 6A**). Further, we detected 47S rRNA signal in cells other than endothelial cells in metaphysis, but the signal was not significantly reduced in the non-endothelial cells upon MPS treatment (**Figure 6B**). Our results also show that in response to GC treatment, vascular endothelial cells primarily undergo senescence, whereas osteoblast lineage may have multiple cell fate changes, such as cell senescence and apoptosis (**Figure S5**) (*Weinstein et al. J Clin Invest 1998; O'Brien et al. Endocrinology 2004; Li et al. PLoS One 2012; Li et al. Cell Death Dis 2013*). Together, our data suggest that vascular endothelial cells are a primary target of ANG and a main cell type undergo cellular senescence in response to GC treatment.

2. Transgenic reporter mouse line, Nestin-GFP was used to label blood vessels in the metaphysis. It marks mesenchymal cells as well. The endothelial-specific expression should be illustrated using Cdh5- iCre driven reporter or any other reporter mice to label bone blood vessels and ANG localisation in type H cells and not in other mesenchymal population.

Response: In the initial submission of this manuscript, we used nestin deficiency as a marker of senescent cells because we previously found that loss of nestin expression represents a feature of

cellular senescence in metaphysis of long bone (*Nat Commun* 2017; 8:1312). In the revised manuscript, we have included a newly established senescence reporter mouse strain $p16^{tdTom}$, in which tandem-dimer Tomato (tdTom) was knocked into exon 1 α of the $p16^{INK4a}$ locus to enable the identification of $p16^{INK4a}$ -activated cells (tdTom⁺) at the single-cell level (*Proc Natl Acad Sci U S A.* 2019; 116: 2603-2611). We have therefore included a new set of data demonstrating the dynamics of cellular senescence at metaphysis in response to GC treatment (**Figure 1A-1F** and **Figure 2A-2F**). The data using Nestin-GFP as a maker has therefore been removed from the revised manuscript.

4. In Fig 2a, is beta-gal localisation nuclear?

Response: SA- β Gal, as a lysosomal enzyme, should not be localized in the nucleus. We have performed SA- β Gal staining with DAPI. Our data showed that SA- β Gal-positive signal is not in nucleus (Figure 1).

Figure 1 Representative images of SA- β Gal staining (blue-green color) (left panel) and DAPI (blue color) (middle panel) in primary spongiosa of femoral bone. In the merged image (right panel), DAPI is switched into pink color in order to distinguish the two different signals.

5. In Fig 3D, both type-H & L cells seem to be reduced. It would be useful to provide quantifications for type-L cells in this experiment and also in transgenic animals (both in the presence and absence of GCs).

Response: As suggested, we have calculated the percentages of type H (CD31^{high}Emcn^{high}) and type L (CD31^{low}Emcn^{low}) blood vessels based on the flow cytometry data. GC treatment induced reductions in the percentage of both vessel subtypes, with the type H vessels being more profoundly affected (**Figure 3D-3F**). The results suggest that GCs suppress the growth of bone blood vessels in general; however, the osteogenesis-coupled type H vessels are the predominant target of GCs. The point has been included in the revised manuscript (Line406-409).

6. Because ANG loss is downstream of GC, whether authors have tested ANG loss of function mouse models to understand they display a bone phenotype similar to GC treated animals.

Response: We recently obtained the $ANG^{flox/flox}$ mice from Dr. Guo-Fu Hu's laboratory and are now generating the osteoclast-specific conditional ANG knockout mice by crossing the $RANK-cre$ mice with the $ANG^{flox/flox}$ mice. As the reviewer pointed out, we expect that this conditional ANG knockout mice develop bone phenotype similar to GC-treated mice. We currently have not obtained the data from the mice yet because it will take quite some time to validate the mouse model and examine the phenotypic changes.

However, our new *in vivo*, *ex vivo*, and *in vitro* data suggest that loss or downregulation of the ANG-PLXNB2 axis contributes to GC-induced cellular senescence. We observed a reduction in ANG⁺ osteoclasts (**Figure 4**) and a simultaneous accumulation of senescent endothelial cells

(**Figure 1-2**) in metaphysis of MPS-treated mice relative to vehicle-treated mice. Moreover, knockdown of PLXNB2 in endothelial cells was sufficient to inhibit ANG nuclear translocation and induce cell senescence in the absence of MPS treatment (**Figure 6**). More importantly, conditioned medium (CM) prepared from the metaphysis explant culture stimulated endothelial cell proliferation and tube formation, which were abolished when an ANG neutralizing antibody or an antibody against ANG-PLXNB2 binding was added in the culture (**Figure 6K**). The results suggest that loss of ANG/ PLXNB2 signaling is sufficient to induce endothelial cell senescence and disrupt the angiogenesis of the cells.

7. Why some experiments were done with DEX, use prednisolone in all. If there is a reason, explain.

Response: The reviewer raises an important point about the different synthetic GCs used in the *in vivo* and *in vitro* studies in our initial submission. We used methylprednisolone (MPS) in all the *in vivo* experiments and dexamethasone (DEX) in the *in vitro* studies. A major reason for the usage of MPS in the mouse study is that the adverse effect of MPS on growing skeleton of young mice has been validated in my laboratory (*Li et al. Nature Communications 2017*) and other groups (*Yang et al. Bone 2018; Peng et al. J Bone Miner Res 2020*). Given that DEX is the most commonly used GC type for *in vitro* studies, we chose DEX in our cell culture study in our original submission. We have recently found that ANG is primarily secreted by RANK⁺TRAP⁺ osteoclasts but not by vascular endothelial cells in metaphysis. We have further revealed that vascular endothelial cells are a main target of ANG rather than serving as a source of it. Therefore, all the *in vitro* data using DEX has been removed in the revised manuscript. Instead, we have conducted a panel of new *ex vivo* and *in vitro* study (**Figure 5 and 6**), in which water-soluble MPS has been used throughout.

8. In fig 4, what is 47S rRNA? The pre-ribosomal RNA is 45S.

Response: The early stage of rRNA production in cells results in the synthesis of the 47S pre-rRNA, which is cleaved at both ends to generate the 45S pre-rRNA and then processed by two alternative pathways to produce mature 18S, 5.8S and 28S rRNAs. We have added more explanation on the rRNA synthesis in the revised manuscript (Line 295-297).

9. No cytoplasmic ANG in figure 5

Response: We are sorry for not providing high-resolution images on the ANG staining in our initial submission. We have obtained a specific ANG antibody that was kindly provided by Dr. Guo-fu Hu's laboratory. In the past year, we have repeated the immunofluorescence staining of femoral bone tissue sections using this antibody. Our new data showed that ANG is almost exclusively expressed in RANK⁺TRAP⁺ osteoclasts in primary spongiosa adjacent to growth plate, and GC treatment reduced the expression of ANG in osteoclasts (**Figure 4**). Please also see our response to the General Comments of this reviewer.

10. What is EU labelling?

Response: 5-ethynyl uridine (EU) is an alkyne-modified nucleoside which is used in the Click-iT[®] RNA Imaging Assay to detect newly synthesized RNA in cells and tissues. Specifically, when the alkyne-containing nucleoside (EU) is fed to cells, it can be actively incorporated into the cells.

Then the modified RNA nucleoside can be detected with a corresponding azide-containing dye and the fluorescence intensity of EU reflects the amount of newly synthesized RNA. We have added detailed explanation on the EU experiment in the Materials and Methods section (Line 697-701) and the Results section (Line 315-317).

11. ANG role in myeloid progenitors has been reported (Goncalves et al., 2016 Cell). The authors demonstrate that osteoclasts are not affected. The reason could be explained.

Response: Previous studies from Hu's laboratory show that ANG is expressed in bone marrow osteolineage cells in close proximity to transplanted hematopoietic stem/progenitor cells (HSPCs) to regulate the function of HSPCs (Silberstein et al. *Cell Stem Cell* 2016; Goncalves et al. *Cell* 2016). We found ANG⁺ cells are almost exclusively expressed in RANK⁺TRAP⁺ osteoclasts in primary spongiosa under the growth plate. A few scattered ANG⁺ cells were also detected in bone marrow of the central part of diaphyseal bone (**Figure 4D** and **4F**). Whether these cells are mesenchymal stem or osteoprogenitor cells remain to be determined. Nevertheless, our results and the findings from Hu's group suggest that the cell source of ANG and its function are distinct in different regions of bone and may also be growth phase-dependent. This important point has been discussed in the revised manuscript (Line 428-436).

12. To understand the endothelial-specific expression pattern of P16 ink4a and HMGB1 in many images, they need to be co-stained with endothelial marker/reporter in all the images.

Response: We recently acquired a murine senescence reporter strain *p16^{tdTom}* (generated by Sharpless group) from Jackson Lab. To determine the endothelial-specific expression pattern of the senescence marker tdTom, we have conducted double-immunofluorescence staining using the bone tissue sections of the *p16^{tdTom}* mice. Our results show that numbers of tdTom-expressing vascular endothelial cells increased in metaphysis in MPS-treated vs. vehicle-treated mice, starting at 1 week, reaching a peak at 2 weeks, and declining at 3 weeks after treatment (**Figures 2A-2F**). Moreover, both double-immunofluorescence staining and flow cytometric analysis of cells in metaphysis of *p16^{tdTom}/+* mice after MPS treatment also showed a strong induction of tdTom expression in vascular endothelial (Emcn⁺) cells (**Figure 2A-2F**).

13. Fig. 7 needs appropriate controls. The phenotype of blood vessels in transgenic mice in the presence and absence of GCs should be shown.

&

14. The vascular phenotype of p16 iKO should be defined.

Response: We agree with the reviewer that the wild-type and transgenic mice in the absence of GC treatment should be included. In the past year, we have added these 2 treatment groups. The changes in cellular senescence, angiogenesis and osteogenesis in metaphysis have been assessed. The new data is shown in new **Figure 3J-3O**.

15. Fig 2h, what is cd14?

Response: We are sorry for the wrong labeling in this figure. "cd14" should be "CD144". We have corrected this error in the revised manuscript.

Response to comments from Reviewer #4:

General Comments:

This manuscript reports that angiogenin insufficiency in bone vascular cells is a reason for glucocorticoids (GC)-induced osteoporosis. Convincing data was presented to show that GC treatment in young mice suppresses nucleolar expression of ANG selectively in endothelial cells comprising type-H vessels in the metaphysis and that ANG-mediated rRNA transcription is an underlying mechanism for endothelial cell senescence. The authors also demonstrated that systemic treatment with ANG rescues GC-induced growth retardation accompanied with normalization of rRNA transcription and endothelial cell proliferation. These data identified a novel GC target with significant therapeutic implications to lessen the adverse effects associated with long term use of GC. My major comment, which is not a concern, is that cell autonomous and non-cell autonomous function of ANG in bone endothelial cells of the spongiosa regions are not clear. The authors presented some evidence that endogenous ANG in endothelial cells played a role but ANG is also secreted by other types of cells such as osteoblasts. Are those ANG, released into the micro-environment, not taken up by endothelial cells to compensate the loss? Are other cells affected by GC in term of ANG secretion? This needs to be addressed, at least discussed, in light of the concern that the conclusion of endothelial cells being the primary GC target cells was not sufficiently supported (see below).

Response: We thank Reviewer #4's encouraging comments and very constructive suggestions. We agree with the reviewer that the cell autonomous and non-cell autonomous function of ANG in bone endothelial cells were not clearly clarified in our initial submission. In the past year, we have conducted a panel of new *in vivo*, *ex vivo*, and *in vitro* experiments, providing evidence for a non-cell autonomous function of ANG in growing skeleton. Specifically, our *in vivo* experiments show that ANG is almost exclusively expressed in RANK⁺TRAP⁺ osteoclasts in metaphysis, particularly in the primary spongiosa adjacent to growth plate. Importantly, GC treatment led to dramatic reductions in both the number of the osteoclasts in this region and the expression of ANG in osteoclasts (**Figure 4B-4J** and **Figure S6**). Moreover, we established an *ex vivo* model system, in which endothelial cells were cultured with conditioned medium (CM) prepared from femoral metaphysis explant culture. Consistent with the *in vivo* finding, our *ex vivo* results show that there was a dramatic reduction in ANG secretion in response to GC treatment. As a result, there were declines in the proliferation and angiogenesis of endothelial cells that were cultured with the CM from MPS-treated explants relative to the CM from control explants (**Figure 5**).

Osteoclasts at the bone/cartilage interface (primary spongiosa) were recently termed as vessel-associated osteoclast subtype with no bone-resorbing activity. This distinct subset of osteoclasts is closely associated with type H vessels in metaphysis and is essential for maintaining angiogenesis and osteogenesis in this region (*Nat Cell Biol* 2019; 21:430-441). Our result that the ANG-expressing osteoclasts are closely associated with Emcn⁺ vascular endothelial cells (**Figure 4K**) agrees with this finding. More importantly, our work provides a molecular mechanism for the osteoclast-vascular crosstalk in this specific skeletal region. Our new *in vivo* and *in vitro* results show that ANG-Plexin B2 (PLXNB2) axis mediates the osteoclast-vascular interplay, which is essential to maintain vascular endothelial cells from senescence in metaphysis of growing bone. Specifically, we found that PLXNB2, a well-recognized ANG receptor, is strongly expressed in

vascular endothelial cells in metaphysis of growing bone (**Figure 6A**). Moreover, knockdown of PLXNB2 in endothelial cells was sufficient to inhibit ANG nuclear translocation and rRNA transcription, resulting in cellular senescence and impaired angiogenic activity (**Figure 6E-6K**). In contrast, administering a recombinant ANG (rh-ANG) significantly attenuated endothelial cell senescence, improved angiogenesis-osteogenesis, and successfully protects growing bone from the negative effects of GC treatment on bone growth and bone formation (**Figure 7**). Collectively, our new results suggest that osteoclast-derived ANG maintains the osteogenesis-coupled type H vessels in metaphysis of growing skeleton via an ANG/PLXNB2-rRNA transcription signaling pathway; and inhibition of this pathway by GCs leads to vascular cell senescence and resultant impaired angiogenesis with osteogenesis (illustrated in **Figure 8**). Because of the critical new findings described above, we have made substantial changes to the manuscript and changed the title of the manuscript into “Osteoclasts Protect Bone Blood Vessels Against Senescence through the Angiogenin/Plexin-B2 Axis”.

Specific Comments:

1. Figure 1J, Nestin positive cells were decreased after MPS treatment. Are those endothelial cells or osteoprogenitor cells? Are endothelial cells in the primary and secondary spongiosa regions of metaphysis bone nestin positive?

Response: Our purpose of employing *Nestin-GFP* in our initial submission was to demonstrate that nestin⁺ cells are highly proliferative cells, and nestin-deficient cells are senescent cells as we previously identify nestin deficiency as a marker of senescent cells in metaphysis of long bone (*Li et al. Nature Communications 2017*). Based on our double-immunofluorescence staining evidence and several other lineage-tracing studies (*Itkin et al. Nature 2016*; *Ono et al. Dev Cell 2014*; *Mendez-Ferrer et al. Nature 2010*), nestin⁺ cells in bone represent a heterogeneous population, with the majority of the nestin⁺ cells being vascular endothelial cells and some nestin⁺ cells being osteoprogenitors. In the revised manuscript, we have removed the study of using *Nestin-GFP* mice to avoid of the cell lineage identity issue. Instead, we have included a newly developed senescence reporter mouse strain *p16^{tdTom}*, using which we found that vascular endothelial cells and osteoprogenitor cells in the metaphysis of growing bone are the main cell lineages that undergo GC-induced senescence in young mice (**Figure 1 and 2**).

2. Figure 1G shows that there is an apparently lower frequency of HMGB1 positive cells upon MPS treatment but the overall intensity of HMGB1 staining is more or less the same in positive cells even though nuclear localization of HMGB1 is lost. However, Figure 2O shows a decrease in HMGB expression in CD144 positive cells but no change in the frequency of positive cells. What is the reason for this discrepancy? Are all HMGB1 positive cells in Figure 1G CD144 positive? Double IF should be done to ensure if only endothelial cells are senescent after MPS treatment. The conclusion that GC treatment primarily causes cellular senescence of vascular endothelial cells in the metaphysis of glowing bone is not sufficiently supported.

Response: We thank the reviewer’s very careful review. We recognize that the HMGB1 expression patterns in response to GC treatment are different in the *in-situ* immunofluorescence staining (**Figure 1J**) and the imaging flow study (**Figure 2N**) based on the calculation methods in our initial submission. The reason for the discrepancy could be that we counted the cells with low

level/dim fluorescence signal (HMGB1^{low/dim} cells) as HMGB1-negative cells in the *in-situ* staining analysis; however, the imaging flow machine counted both HMGB1^{high/bright} cells and HMGB1^{low/dim} cells as positive in the single cell flow data. In the revised manuscript, we have recalculated the data using only fluorescence intensity as a parameter in both *in-situ* staining (**Figure 1K** and **1L**) and the single cell flow (**Figure 2Q** and **2T**). We believe that this way of calculation can accurately reflect the change of HMGB1 expression.

To validate the senescence phenotype of bone vascular cells in response to GC treatment, we have included a murine senescence reporter strain *p16^{tdTom}*. Using this mouse strain, we have identified that vascular endothelial cells and osteoprogenitor cells are main cell lineages undergoing senescence as detected by co-immunofluorescence staining and flow cytometry analysis (**Figure 2A-2F**). Of note, the number of tdTom-expressing cells increased nearly 6-fold in the *Emcn*+ cell population but only to 3-fold in the *Osx*+ cell population at the peak (Figure 2B and 2D), suggesting that vascular endothelial cells are more susceptible than osteoprogenitor cells to MPS treatment. Supporting this assumption, we also show that most vascular endothelial cells (approximately 83.43±4.57%) have PLXNB2 expression, but only a small portion of *Osx*+ osteoprogenitor cells (approximately 5.76±3.89%) are PLXNB2-positive (**Figure 6A**). The FISH assay of bone tissue sections also shows that vascular endothelial cells in metaphysis have strong 47S rRNA signal in vehicle-treated mice but significant reduced after MPS treatment. However, the 47S rRNA signal in the non-endothelial cells was not significantly reduced upon MPS treatment (**Figure 6B-6D**). Further, GC treatment also stimulates cell apoptosis in osteoprogenitor cells but not in endothelial cells (**Figure S4**). Therefore, in response to GC treatment, vascular endothelial cells primarily undergo senescence, whereas osteoblast lineage likely has multiple cell fate changes, such as cell apoptosis, cell senescence, or lineage shift (*Weinstein et al. J Clin Invest 1998; O'Brien et al. Endocrinology 2004; Li et al. PLoS One 2012; Li et al. Cell Death Dis 2013*). Together, our data suggest that vascular endothelial cells are a primary target of ANG and a main cell type undergo cellular senescence in response to GC treatment.

3. Figure 3. Do you see any difference in L-type blood vessels?

Response: We have calculated the percentages of type H (CD31^{high}*Emcn*^{high}) and type L (CD31^{low}*Emcn*^{low}) blood vessels based on the flow cytometry data. GC treatment induced reductions in the percentage of both vessel subtypes, with the type H vessels being more profoundly affected (**Figure 3D-3F**). The results suggest that GCs suppress the growth of bone blood vessels in general; however, the osteogenesis-coupled type H vessels are the predominant target of GCs. The point has been included in the revised manuscript (Line406-409).

4. Fig. 5G. How many ANG-specific siRNA were tested? Do you have mRNA and protein data on the knockdown efficiency?

Response: Since the non-cell autonomous function of ANG has been identified in our revised manuscript, the study using ANG-siRNA in endothelial cells has been removed. Instead, we have used PLXNB2 siRNA and examined the changes of endothelial cells after knocking down Plexin B2. We have tested 2 PLXNB2 siRNA, and protein data showing the knockdown efficiency has been presented (**Figure 6E**).

5. Figure 8H and 8I. In the text and figure legend, Figure 8H and 8I are supposed to be *Osx* positive cells, however, the figures actually show beta-Gal positive vessels.

Response: The error has been corrected in the revised text and figure legend.

6. Page 18, Gloria Su appeared to be a coauthor and should not be acknowledged for providing the mice, it will be more appropriate to give a citation of these mice.

Response: The information has been removed from the Acknowledgement section. A citation regarding the *p16* flox mice has been included (Line497-499).

7. The ANG product used in systemic administration was a 15 amino acid synthetic peptide near the center of human ANG (Thermo Fisher, PEP-1464, page 19, line 5 from bottom) rather than the entire ANG protein. If the authors have not examined the activity of recombinant ANG protein, it should be done before the conclusion that it is the ANG-ribosome biogenesis pathway that is responsible for the therapeutic activity of ANG against GC-induced osteoporosis. It is hard to believe that a 15 amino acid peptide of ANG from the center part of the protein will restore ribosome biogenesis. I hope that it is only a typo when the corresponding author wrote the manuscript.

Response: We sincerely appreciate the reviewer's carefulness, and yes, this is a typo of the corresponding author. In fact, we used the 15 amino acid ANG peptide (Thermo Fisher, PEP-1464) as a blocking peptide in confirming the specificity of one of the ANG antibodies. The recombinant ANG protein used for the *in vivo* study (**Figure 7**) was purchased from R & D system (catalog number: 265-AN-250). The information has been corrected in the revised manuscript.

8. Page 20, line 8. Thermo Fisher 4390843 is a control siRNA not ANG-specific siRNA, please correct.

Response: We are sorry for the wrong description of the control siRNA and ANG-specific siRNA. We have deleted the information since the data using siRNA for ANG has been removed from the revised manuscript.

REVIEWER COMMENTS

Reviewer #1 (Remarks to the Author):

This is a very interesting concept that growth retardation in young growing mice is dependent on vessel senescence and that there is a cross-talk between osteoclasts and vasculature. The manuscript underwent a very extensive revision and many new experiments including the use of new mouse line addressed the concerns of reviewers. For me only a few items remains to be solved.

The direct effect of GCs on osteoclast to regulate Ang expression is still not convincingly shown. The use of RU486 in metaphyseal cultures after 3 days resembles that overall GR is involved in the regulation of expression. But whether this is a direct target is not clear from the timing, and the origin of cells is formally not shown. This could be accomplished easily. Of course it could be that Ang is not directly regulated by GR.

According to the point above, I appreciate that the authors determined the absolute numbers of osteoclasts. However, I can not follow the line of evidence that Ang is reduced within the osteoclasts. In this context it would be interesting to note whether the high fraction of RANK+ and TRAP+ cells expressing Ang remains the same after MPS treatment.

For the adult mice and at least for the experiment shown in Figure 3 and 7 classical histomorphometry due to the accepted rules should be performed. For GC induced effects inhibition of bone formation is a hall mark, and the death of osteocytes. I appreciate that the numbers of osteocytes had been had been addressed in the rhANG administration experiment. In case dynamic histomorphometry can not be done, since for obvious reasons fluorescent dies were not applied this should be accomplished by determination of P1NP in serum.

Analysis of the vertebrae with classical bone phenotyping methods (microCT, histomorphometry) would be desirable.

The authors could consider that the conditioned medium experiments could be added with a neutralizing antibody to show that Ang in the CM is indeed doing the job.

Reviewer #2 (Remarks to the Author):

The authors have done an excellent job addressing my concerns. The new data with the p16 reporter mice is convincing and they have also addressed my other points raised.
Sundeep Khosla

Reviewer #3 (Remarks to the Author):

Authors have thoroughly addressed all the comments, modified the manuscript and removed the data which were not fitting. Some figures need reorganisation to follow the text eg. figures 3,7.

Reviewer #4 (Remarks to the Author):

The authors have addressed all of my concerns and comments with new experiments. I have no other concerns and I recommend acceptance of this manuscript.

Point-by-point response to reviewers' comments

We appreciate all the constructive comments on the revised version of our manuscript and are pleased to know that Reviewer #2 and #4 consider our work ready for publication. Reviewer #3 acknowledges that “we have thoroughly addressed all the comments” but “Some figures need reorganization” prior to publication. As suggested, we have reorganized Figure 3 and 7 in the revised manuscript. Reviewer #1 suggested us to further evaluate the direct effect of GCs on osteoclasts for Ang expression and perform bone histomorphometry analysis in the *in vivo* studies. The following responses have been prepared to address Reviewer #1' s comments in a point-by-point fashion.

Response to comments from Reviewer #1:

This is a very interesting concept that growth retardation in young growing mice is dependent on vessel senescence and that there is a crosstalk between osteoclasts and vasculature. The manuscript underwent a very extensive revision and many new experiments including the use of new mouse line addressed the concerns of reviewers. For me only a few items remain to be solved.

1. The direct effect of GCs on osteoclast to regulate Ang expression is still not convincingly shown. The use of RU486 in metaphyseal cultures after 3 days resembles that overall GR is involved in the regulation of expression. But whether this is a direct target is not clear from the timing, and the origin of cells is formally not shown. This could be accomplished easily. Of course, it could be that Ang is not directly regulated by GR. According to the point above, I appreciate that the authors determined the absolute numbers of osteoclasts. However, I cannot follow the line of evidence that Ang is reduced within the osteoclasts. In this context it would be interesting to note whether the high fraction of RANK⁺ and TRAP⁺ cells expressing Ang remains the same after MPS treatment.

Response: We appreciate this constructive comment. We have separately calculated the number of total osteoclasts and the number of osteoclasts expressing ANG. Our results show that both the total number of osteoclasts and the number of osteoclasts expressing ANG were significantly reduced in response to MPS treatment (Figure 4H and 4K). However, the percentages of ANG-expressing cells in the RANK⁺ and TRAP⁺ cell populations remain the same in the methylprednisolone (MPS)-treated group relative to vehicle-treated group (Figure 4I and 4L). Therefore, the reviewer is correct that glucocorticoids (GCs) primarily induce the diminishment of the ANG-expressing osteoclasts. Our results from the *ex vivo* metaphysis explant culture experiment, showing that osteoclast stimulator PTH increased the ANG production and osteoclast inhibitor alendronate decreased ANG production, support this assumption.

To further clarify whether GCs may directly regulate ANG gene expression in osteoclasts, we have performed *in vitro* osteoclastogenesis assays. Our results show that many TRAP⁺ multinuclear osteoclasts were formed from bone marrow macrophages (BMMs) at 5 days after M-CSF and RANKL treatment. Multinuclear osteoclasts were markedly reduced when MPS was concomitantly added with M-CSF and RANKL at the beginning of the culture (Figure 5Ai, 5Bi, and 5Ci). This result is consistent with previous findings that GCs inhibit the proliferation of

osteoclast precursors and osteoclast spreading (Kim et al., *JCI* 2006; 116:2152-2160 and Kim et al., *Ann. N. Y. Acad. Sci.* 2007; 1116:335-9). Of note, ANG expression had 4-fold reduction in MPS-treated cells (primarily mononuclear preosteoclasts) relative to vehicle-treated cells (mature osteoclasts) (Figure 5Di). While the number of mature osteoclasts was restored by addition of GR antagonist RU486, ANG expression in these cells was also elevated. The results suggest that mature osteoclasts are a main source of ANG. In another set of experiments, MPS was added at the end stage when multinuclear osteoclasts were already formed (Figure 5Aii). The number of mature osteoclasts was not significantly changed in MPS-treated vs. vehicle-treated cells (Figure 5Bii and 5Cii); and importantly, the difference in ANG expression levels in these two treatment groups was also not detected (Figure 5Dii). This new set of *in vitro* data, together with our *in vivo* and *ex vivo* results, suggest that GC treatment inhibits ANG production in metaphysis of long bone primarily by inducing the diminishment of ANG-expressing osteoclasts rather than by directly regulating ANG gene expression. The new results and discussion on this point has been added in revised manuscript (page 11, lines 239–243; page 12, lines 255–269; and page 20-21, lines 468–487).

2. For the adult mice and at least for the experiment shown in Figure 3 and 7 classical histomorphometry due to the accepted rules should be performed. For GC induced effects inhibition of bone formation is a hall mark, and the death of osteocytes. I appreciate that the numbers of osteocytes had been addressed in the rhANG administration experiment. In case dynamic histomorphometry cannot be done, since for obvious reasons fluorescent dies were not applied this should be accomplished by determination of P1NP in serum.

Response: As suggested, we have now included bone histomorphometry data in both experiments. Specifically, in the rhANG rescue experiment (Figure 8), we actually injected the double-labeling dye into mice and saved all the bone samples. We did not include the histomorphometry measurements in our original submission due to the overlength of the manuscript. Our new results show that MPS treatment reduced osteoblast number and surface (Figure S9A-S9C) as well as the number of osteocytes (Figure S9D and S9E), which were all largely rectified by rhANG co-treatment. Improved bone formation by rhANG was also confirmed by Goldner's Trichrome staining and double calcein labeling analysis. Femoral metaphysis of MPS-treated mice showed a decrease in newly formed bone (red), which was improved by rhANG co-treatment (Figure 8M and 8N). Double calcein labeling showed that the distance between two consecutive labels was less in MPS-treated mice than in the vehicle-treated control mice and became more in rhANG-cotreated mice (Figure 8O). The reduced BFR/BS in response to MPS treatment was improved by rhANG co-treatment (Figure 8P). These results agree with the data included in our original submission that rhANG has significant effects on alleviating GC-induced impairment on bone formation. Moreover, histomorphometry analysis in the experiment using the inducible p16 knockout mice reveal that MPS treatment-induced reductions in osteoblast number and surface as well as serum P1NP, a marker for bone formation, were largely improved in p16 iKO mice vs. WT littermates. The new data has been included in Figure S6.

We acknowledge that exogenous GCs have divergent effects on osteoclasts both *in vivo* and *in vitro* due to differences in strain, age, glucocorticoid dose, or experimental set-up. Our finding that MPS treatment decreases osteoclast numbers in femoral metaphysis in growing young mice is consistent with previous work conducted using the same age of young mice (Peng et al., *JBMR*

2020; 35:1188-1202 and Yang et al., *Bone*, 2018. 114: 1-13). GC treatment has been shown to result in increased osteoclasts in adult skeleton (Sato et al., *JBMR* 2016; 31:1791-802 and Sato et al., *Endocrinology* 2019; 160:1659-73). The different effects on osteoclasts exerted by GCs is likely attributable to the differences in the age of the mice, the bone region for analysis, and the dosage of GCs.

3. Analysis of the vertebrae with classical bone phenotyping methods (microCT, histomorphometry) would be desirable.

Response: We appreciate this suggestion. Our study focuses on the role of metaphyseal blood vessel senescence in mediating the adverse effects of GCs on growing long bone. While it is interesting to investigate the changes of the skeleton at other locations in our mouse models, it can take quite a long time to get a clear answer with regard to whether the adverse effects of GCs on vertebrae in young mice are mediated via the same mechanism as it did on long bone. It is very important to note that GC treatment also induces bone loss in lumbar vertebrae in adult mice (Sato et al., *JBMR* 2016; 31:1791-802; Sato et al., *Endocrinology* 2017; 158: 664–677; and Sato et al., *Endocrinology* 2019; 160:1659-73). We will further examine the role of osteoclast-secreted ANG in mediating the action of GCs on lumbar vertebrae of young mice as our next project. We have included discussion on this point in the revised manuscript (page 23, lines 526–532).

4. The authors could consider that the conditioned medium experiments could be added with a neutralizing antibody to show that Ang in the CM is indeed doing the job.

Response: This is an important experiment that was already included in the last version of our manuscript. In Figure 7K and 7L, we took advantage of an ANG neutralizing antibody (26-2F) and an antibody specifically blocking the ANG-PLXNB2 binding (mAB17) and conducted the *ex vivo* tube formation assays. Both antibodies completely abolished angiogenic activity induced by the conditioned medium prepared from metaphysis explant culture.

REVIEWERS' COMMENTS

Reviewer #1 (Remarks to the Author):

The authors addressed the remaining concerns. Congratulations to this study.

Point-by-point response to reviewers' comments

We appreciate all reviewers' constructive comments on the revised version of our manuscript and glad to see that Reviewer #1 consider our work ready for publication.